

# A New Differential Optical Absorption Spectroscopy Instrument to Study Atmospheric Chemistry from a High-Altitude Unmanned Aircraft

Jochen Stutz[1], Bodo Werner[2], Max Spolaor[1], Lisa Scalone[2], James Festa[1], Catalina Tsai[1], Ross Cheung[1], Santo F. Colosimo[1], Ugo Tricoli[2], Rasmus Raecke[2], Ryan Hossaini[3], Martyn P. Chipperfield[4], Wuhu Feng[4,5], Ru-Shan Gao[6], Eric J. Hintsa[6,7], James W. Elkins[6], Fred L. Moore[6,7], Bruce Daube[8], Jasna Pittman[8], Steven Wofsy[8], and Klaus Pfeilsticker[2]

[1]Department of Atmospheric and Oceanic Science, University of California Los Angeles, Los Angeles, California, USA
[2]Institute of Environmental Physics, University of Heidelberg, Heidelberg, Germany
[3]Lancaster Environment Centre, University of Lancaster, Lancaster, UK
[4]Institute for Climate and Atmospheric Science, School of Earth and Environment, University of Leeds, Leeds, UK
[5]National Centre for Atmospheric Science, School of Earth and Environment, University of Leeds, UK
[6]NOAA Earth System Research Laboratory, Boulder, Colorado, USA
[7]Cooperative Institute for Research in the Environmental Sciences (CIRES), University of Colorado, Boulder, Colorado, USA
[8]School of Engineering and Applied Sciences, Harvard University, Cambridge, Massachusetts, USA

*Correspondence to:* J. Stutz (jochen@atmos.ucla.edu)

**Abstract.**

Observations of atmospheric trace gases in the tropical upper troposphere (UT), tropical tropopause layer (TTL) and lower stratosphere (LS) require dedicated measurement platforms and instrumentation. Here we present a new limb-scanning Differential Optical Absorption Spectroscopy (DOAS) instrument developed for NASA's Global Hawk unmanned aerial system

5  (GH) during the Airborne Tropical TRopopause EXperiment (ATTREX). The mini-DOAS system is designed for automatic operation under unpressurized and unheated conditions at $14 - 18$ km altitude, collecting scattered sunlight in three wavelength windows: UV ($301 - 387$ nm), visible ($410 - 525$ nm) and near infrared ($900 - 1700$ nm). A telescope scanning unit allows selection of a viewing angle around the limb, as well as real-time correction of the aircraft pitch. Due to the high altitude, solar reference spectra are measured using diffusors and direct sunlight. The DOAS approach allows retrieval of slant column

10  densities (SCD) of $O_3$, $O_4$, $NO_2$, and $BrO$ with relative errors similar to other aircraft DOAS systems. Radiative transfer considerations show that the retrieval of trace gas mixing ratios from the observed SCD based on $O_4$ observations, the most common approach for DOAS measurements, is inadequate for high-altitude observations, due to frequent presence of low altitude clouds. A newly developed technique that constrains the radiative transfer (RT) by comparing in-situ and DOAS $O_3$ observations overcomes this issue. Extensive sensitivity calculations show that the novel $O_3$-scaling technique allows the re-

15  trieval of $BrO$ and $NO_2$ mixing ratios at high accuracies of $0.3 - 0.6$ ppt and 15 ppt, respectively. The $BrO$ and $NO_2$ mixing ratios and vertical profiles observed during ATTREX thus provide new insights into ozone and halogen chemistry in the UT, TTL, and LS.





# 1 Introduction

Transport and transformation of tropospheric gases in the tropical upper troposphere (UT), tropical tropopause layer (TTL), and lower stratosphere (LS) play an important role in controlling stratospheric water vapor and ozone, as well as the formation of thin cirrus clouds and hence the radiative forcing in the TTL (e.g. Fueglistaler et al. (2009)). However, many of the physical

and chemical processes controlling the composition of the UT, TTL and LS remain inadequately quantified, in part due to a lack of accurate observations. This includes the budget of ozone within the TTL and LS, which suffers from a lack of quantitative constraints of the underlying chemical mechanisms. Generally, ozone concentration in the TTL and LS is controlled by its chemical formation and destruction through catalytic cycles involving hydroxyl radicals, reactive bromine species, and, indirectly, nitrogen oxides (Salawitch, 2005). While the significance of reactive halogens for ozone chemistry is generally ac-

cepted, the speciation, transport, chemistry, and total levels of bromine species, $Br_y$ (=$Br_y^{org}$ + $Br_y^{inorg}$), in the TTL are still under debate (WMO, 2015). Long-lived organic bromine species, emitted by natural and anthropogenic sources, constitute the largest portion of the $Br_y$ flux through the TTL. Currently, $CH_3Br$ contributes about 6.9 ppt to $Br_y^{org}$, while halons, such as $CClBrF_2$, $CBrF_3$, $CBr_2F_2$, etc., contribute approximately 8 ppt to $Br_y^{org}$. It is now recognized that very short-lived organic species (VSLS), i.e. species with atmospheric residence times of less than 6 months, and inorganic bromine species ($Br_y^{inorg}$)

contribute to total $Br_y$ in the UTLS. However, their exact contribution still has high uncertainties. It is currently believed that these two sources contribute in the range of 2 - 8 ppt to the stratospheric bromine budget. This high uncertainty partly stems from a lack of simultaneous measurements of VSLS and inorganic bromine species in the TTL (WMO, 2015).

Another difficulty in assessing the role of inorganic bromine species in the UTLS is the inability to measure all inorganic bromine species. Most knowledge on the presence of inorganic bromine has been derived from observations of BrO (Harder

et al., 1998; Ferlemann et al., 1998; Pfeilsticker et al., 2000; Fitzenberger et al., 2000; Pundt et al., 2002; Weidner et al., 2005; Dorf et al., 2006, 2008). However, the contribution of BrO to inorganic bromine, $Br_y^{inorg}$, is influenced by other species, in particular ozone and $NO_2$. Consequently, these species have to be measured simultaneously to allow an accurate quantification of the contribution of inorganic bromine to $Br_y$. Because the BrO/$Br_y^{inorg}$ ratio is typically in the range of 10% - 90%, largely depending on ozone and nitrogen oxide levels (Fernandez et al., 2014; Navarro et al., 2015; Werner et al., 2016), this calculation

imposes high requirements for the accuracy of BrO observations. To further improve our knowledge of the $Br_y$ budget, BrO has to be known to better than 1 ppt to reduce the current uncertainty on $Br_y$ levels of ±3 ppt. In addition, $NO_2$ and ozone have to be determined at the same time and at high accuracies as well.

The investigation of the TTL has historically been challenging because altitudes between 14 – 20 km are difficult to reach with most measurement platforms. Ozone levels are fairly well known due to regular balloon-borne observations. Other mea-

surements, in particular of reactive trace gases, are rather sparse (WMO, 2015). Observations using high-altitude balloons have been common for studying polar stratospheric ozone chemistry. The earliest measurements were performed using in-situ chemical conversion resonance fluorescence BrO instruments (Brune et al., 1988; Woyke et al., 1999). In recent years, however, balloon-borne BrO and $NO_2$ (and IO) measurements have been based on remote sensing instruments using solar occultation UV-vis absorption spectroscopy in the limb (Harder et al., 1998; Ferlemann et al., 1998; Fitzenberger et al., 2000; Pfeilsticker



et al., 2000; Pundt et al., 2002; Weidner et al., 2005; Dorf et al., 2006, 2008; Weidner et al., 2005; Kritten et al., 2010). The advantage of this method is its high sensitivity due to very long absorption paths, and thus low detection limits of the path-integrated BrO and $NO_2$ concentrations, i.e. slant column densities (SCD), in the range of $5 \times 10^{12}$ molec./cm$^2$ for BrO (Harder et al., 1998; Ferlemann et al., 1998; Fitzenberger et al., 2000) and $1 \times 10^{15}$ molec./cm$^2$ for $NO_2$ (Butz et al., 2006;

Kritten et al., 2010). Considering the uncertainties in the RT-based retrievals, BrO mixing ratio uncertainties are in the range of 0.5 – 2 ppt, while those of $NO_2$ are around 30 ppt (Weidner et al., 2005; Butz et al., 2006; Kritten et al., 2010). While much of our current understanding of stratospheric BrO has been derived from these observations, the required balloon launches are challenging and expensive and, consequently, are performed infrequently. In addition, balloon-borne observations offer little information on the horizontal distribution of the observed trace gases. It is thus not surprising that only a few balloon borne

BrO observations in the TTL have thus far been reported (Pundt et al., 2002; Dorf et al., 2006, 2008).

While satellite observations of stratospheric $NO_2$ have been made for several decades, for example by the various SAGE instruments (Chu and McCormick, 1986; Polyakov et al., 2005), BrO observations from space have only been available since the launch of the Global Ozone Monitoring Experiment, GOME, in 1995 (Burrows et al., 1999). The GOME instrument, and many later NADIR instruments, are able to separate tropospheric and stratospheric VCD, but provide no information on the

UTLS. The early limb scanning systems, such as SAGE, did not have the sensitivity to accurately measure $NO_2$ in the UTLS, and were not able to observe BrO. Only modern limb-scanning systems are capable of providing the vertical resolution and sensitivity required to study the chemistry in the UTLS. For example, the limb mode of the SCIAMACHY instrument achieves a vertical resolution of 3.5 – 6 km in the 14 – 24 km altitude range, with BrO uncertainties of 2 – 4 ppt (Rozanov et al., 2011; Parrella et al., 2013) and an $NO_2$ uncertainty of around 30 ppt (Bauer et al., 2012). The Odin Optical Spectrograph and

Infrared Imager System (Odin/OSIRIS) achieves a vertical resolution of 2 - 3 km and similar detection limits (e.g. Haley et al. (2004); McLinden et al. (2010)). The Microwave Limb Sounder, MLS, also provides vertical BrO profiles, although spatial or temporal averages have to be used to reduce uncertainties to sufficiently low mixing ratios to allow interpretation to the data (Millán et al., 2012; Stachnik et al., 2013).

Few research aircraft are able to reach altitudes above 14 km. Exceptions are NASA's ER2 (e.g. Wennberg et al. (1998)), and

more modern jets such as NSF's HIAPER (e.g. Volkamer et al. (2015)). In-situ $NO_2$ observations, for example using photolytic conversion / chemiluminescence, have been well established on a number of aircraft, including the high altitude aircraft (e.g. Gao et al. (1994); Ryerson et al. (2000)). BrO observations using the chemical conversion resonance fluorescence on board the NASA ER2 yielded substantial insights into the role of bromine in stratospheric ozone, Brune et al. (1988); Toohey et al. (1990) . Despite a high sensitivity of ∼1 ppt, this instrument has not been used in the past two decades, and no measurements

in the tropics have been reported. Most recently, airborne chemical ionization mass spectrometer observations of BrO have become available (Neuman et al., 2010), but have thus far only been used for measurements in the lower and free troposphere.

Aircraft remote sensing of BrO and $NO_2$ has become popular in recent years. While some of the early observations have focused on column abundances of BrO (Wahner et al., 1990; Erle et al., 1998), newer instruments have been used to derive vertical concentration profiles (Bruns et al., 2004; Prados-Roman et al., 2011; Baidair et al., 2013; Volkamer et al., 2015).

All of these observations rely on the measurement of the solar scattered light sampled in the limb, or a combination of other



elevation angles (EA). The trace gas absorptions in the measured solar radiance spectrum are analyzed by the Differential Optical Absorption Spectroscopy, DOAS, technique (Platt and Stutz, 2008). In short, DOAS is based on the identification and quantification of narrow-band trace gas absorption structures in the UV-visible wavelength range, using known absorption cross sections and least squares fitting procedures. The retrieval typically yields a concentration integrated over the photon light paths

between the sun and the instrument, the so-called slant column density, SCD. To accurately describe the solar Fraunhofer lines, the analysis is often performed relative to a solar reference spectrum, $SCD_{ref}$, measured with the same instrument, but under conditions with lower trace gas absorptions. The result of a DOAS retrieval is thus commonly referred to as the differential slant column density:

$$DSCD = SCD - SCD_{ref} \tag{1}$$

The main challenge in the interpretation of aircraft DOAS observations is the conversion of DSCD to trace gas concentration profiles. Typically this is done in a two step process. The first step is the derivation of an aerosol extinction profile which allows the description of the RT in the atmosphere at the time of the measurement. Different approaches have been reported in literature, all of which rely on a combination of a radiative transfer model (RTM) and an optimal estimation technique (Rodgers, 2000), which for this non-linear problem is typically iterative. In the second step, the RTM is constrained by the

aerosol extinction profile, in order to derive trace gas concentration profiles, again using optimal estimation (Rodgers, 2000). The setup of the retrievals depends on the selected viewing strategy, as well as on the observed parameter used to derive the aerosol extinction profile. The most common parameter used in passive DOAS application is the oxygen collisional complex, $O_4$, which has a well-known and temporally constant atmospheric profile (Platt and Stutz, 2008).

Theoretical consideration to derive trace gas profiles based on DOAS observations at various EA from an aircraft flying at

one altitude (Bruns et al., 2004) find that vertical resolution of profiles near flight altitude can reach 2 km. However, the study also points out that, above 14 km, the profile information content is small. This approach was applied to measure $NO_2$ vertical profiles (Bruns et al., 2006) at 10 km flight altitude, showing that vertical profiles throughout the troposphere can indeed be retrieved by using four different EA and three different wavelengths. A different approach to retrieve trace gas profiles relies on the capability of the aircraft to ascend and descend in the atmosphere, together with observations in the limb, which provide

the highest sensitivity at flight altitude (Prados-Roman et al., 2011; Baidair et al., 2013; Volkamer et al., 2015). Prados-Roman et al. (2011) describe an approach in which relative radiances from the ascent of the aircraft are used to derive aerosol extinction profiles. A comparison of observed $O_4$ DSCD with those calculated by a RTM, constrained by the derived extinction profiles, shows the applicability of this approach (Prados-Roman et al., 2011). Baidair et al. (2013) and Volkamer et al. (2015) describe a somewhat different approach in which measured $O_4$ DSCD profiles are used to derive the aerosol extinction profile. In both

cases, vertical resolution of the retrieved trace gas profiles are in the range of 0.5 – 2 km throughout the range of the aircraft ascent or descent maneuver.

A lack of accurate measurements has thus far limited our ability to study halogen and ozone chemistry in the TTL. There is thus a need to provide observations in the TTL to supplement the sparse data set on its chemical composition, and to provide, for the first time, observations over the western Pacific. Unmanned aerial systems (UAS) open new frontiers for the study of





the atmosphere. In particular, NASA's Global Hawk, which allows an unprecedented endurance of 25 hours in combination with a ceiling altitude of 20 km, equipped with high quality in-situ and remote sensing instruments, opens the door to study the TTL. However, the use of a high altitude UAS brings with it new challenges, both in design and operation of the instrument, as well as in the interpretation of the observations.

5    Here we present a novel DOAS instrument specifically developed for NASA's Global Hawk UAS to study the chemistry and physics of the TTL. We describe the technical and operational details of the new instrument (Section 2) and the DOAS analysis approach (Section 3). The particular RT condition, and the trace gas retrievals are discussed in Section 4. Examples of the data retrieved during the Airborne Tropical TRopopause EXperiment (ATTREX) 2013 experiment are presented in Section 5. The scientific results of the ATTREX 2013 deployment are discussed in a companion manuscript by Werner et al. (2016).

## 2    Instrument Description

The mini-DOAS instrument used in this study is a custom-built limb-scanning system that is specifically designed for operation in a long-range, high-altitude airframe. The design is based on earlier balloon-borne and airborne mini-DOAS instruments (Ferlemann et al., 1998; Fitzenberger et al., 2000; Weidner et al., 2005; Prados-Roman et al., 2011) and ground-based MAX-DOAS systems (Pikelnaya et al., 2007; Platt and Stutz, 2008). The unique Global Hawk platform (Section 2.1) led to novel opportunities, but also to specific requirements which are described in Section 2.1. The instrument (Figure 1) consists of a telescope/scanner unit (Section 2.2) connected via fiber bundles to a three spectrometer assembly containing UV, visible and near-IR channels (Section 2.3), and supportive thermal control, communication, and house-keeping and data acquisition (Section 2.4).

### 2.1    High Altitude Platform

NASA's Global Hawk (GH) is a high altitude, long-range unmanned UAS used for atmospheric research (http://www.nasa.gov/centers/armstrong/aircraft/GlobalHawk). The GH measures 13.5 m in length and has a wingspan of 35 m. It can support payloads up to ~900 kg. Typical flight altitudes are between 15 and 20 km, although during ATTREX the ceiling was most often limited to ~18 km. Flight duration with a full payload is up to 25 hours, giving the GH a range of ~18000 km at typical flight velocities of 170 m/s. The aircraft is controlled by a ground station via satellite communication.

The comprehensive multi-instrument payload during the ATTREX experiment (Jensen et al., 2013, 2015), the high-altitude and long endurance, and the unpressurized and unheated instrument bay pose specific requirements on the mini-DOAS instrument:

– Size limitation to a cube of ~45cm side length and a total weight less than 35 kg.

– Stable operation in an atmospheric pressure range from ~1013 hPa to ~70 hPa and a temperature range between -80 to +35°C for up to 30 hours.

– Fully automated operation, but with the ability to remotely control the instrument.



– Perform DOAS type measurements at 14 – 18 km altitude, where a zenith scan cannot be used as a solar reference.

– High accuracy to detect small trace gas concentrations in the TTL.

We address the solutions to these requirements in the following section. Figure 1 and Table 1 give an overview of the mechanical and electrical instrument characteristics.

## 2.2 Spectrometer Assembly

The performance of a DOAS instrument depends strongly on the stability of the spectroscopic components (Ferlemann et al., 2000; Weidner et al., 2005; Platt and Stutz, 2008). In particular, for the highly variable environmental conditions on the GH, pressure and temperature stabilization is crucial to achieve the high sensitivity required for TTL observations. Based on previous experience, we use a vacuum vessel to house three optical spectrometers (Figure 1). The vessel is evacuated before every field deployment and, if needed, once or twice during field deployments. The highly varying temperatures, low air density, which inhibits heat transport from heat sinks, and the short time of power availability before take-off, makes the use of an electronic temperature stabilization scheme unfeasible. Thermal stabilization is thus achieved via 9 liters of a water-ice mixture (Figure 1). The ice-water tank and vacuum vessel are insulated with aircraft-quality foam. Spectrometer temperatures during flight are maintained to within $\pm 0.1°$C.

Three symmetric crossed Czerny-Turner grating spectrometers are mounted in the vacuum vessel, covering ultraviolet (UV), visible (VIS) and near-infrared (near-IR) wavelength regions (Table 2). The UV- and VIS-spectrometers are Ocean Optics QE65000 spectrometers with Hamamatsu S7031-1006 CCD detectors. The UV spectrometer has a 2400 groove/mm holographic grating and a wavelength range of 301 - 387 nm, whereas the VIS spectrometer has a 1800 groove/mm holographic grating and a range of 410 - 525 nm. The near-IR spectrometer is an Ocean Optics NIRQuest 512 spectrometer with a Hamamatsu G9204-512 InGaAs linear image sensor and a 150 grove/mm grating, and a wavelength range of 896 - 1730 nm. A 200 $\mu m$ entrance slit is used for the UV, while a 100 $\mu m$ slit is used for the VIS and near-IR . The fibers at the spectrometer end of the bundle are linearly arranged (Table 2) and placed at the entrance slit. It should be noted that a slight change in spectral resolution is present as the instrument is cooled from room temperature to 0°C. The alignment of the fibers is thus performed at low temperatures.

The spectral characteristics, i.e. wavelength pixel mapping and instrument function, of the spectrometers are determined before each flight at 0°C using Hg, Kr, and Ar atomic emission lamps (http:\\www.nist.gov). The spectrometer characteristics are also confirmed using the Fraunhofer lines in the solar reference spectra. Typical results for dispersion and spectral resolution are listed in Table 2.

The Ocean Optics spectrometers are known to have a slight detector non-linearity. To overcome this problem the spectrometer detectors are operated at the same saturation levels (50%) through an adjustment of the integration time. This ensures that non-linearities do not impact the spectral analysis.





## 2.3 Telescopes

Each of the spectrometers is connected to a small telescope/scanning unit using 1.5 m long, NA = 0.22, quartz (UV) and glass fiber bundles (VIS & near-IR) (Table 2). The bundles are arranged linearly on both ends and aligned with the spectrometer entrance slit. On the telescope end, the linear array of fibers is mounted horizontally.

Each telescope consists of a fiber holder, which allows adjustment of the fiber position, and a focusing lens (f = 35 mm, ∅ = 12 mm) made of fused silica for the UV channel and BK7 glass in the visible and near-IR channels. To minimize the UV spectrometer straylight, a HOYA U-340 (∅ = 12 mm) optical bandpass filter is placed in front of the lens of the UV telescope. To suppress the second grating order, a Schott RG 830 optical filter is used for the near-IR channel. No filter is used in the visible telescope.

The final optical components of the telescopes are a 12 mm UV grade fused silica (UV), and uncoated N-BK7 (VIS & near-IR) total internal reflection prisms. These prisms are mounted with one side in flight direction (for a limb geometry) and turn the incoming light beam by 90 degrees onto the filter/lens. Optically the telescopes view a "rectangle" in the sky that is ~0.25° in elevation and ~1° in the azimuth. The elevation opening angles are precisely determined by using the scanning capability of the telescopes.

The prisms are mounted such that they can be rotated in the telescope axis, i.e. approximately perpendicular to flight direction, by a small high precision servo-motor with planetary gear (Faulhaber 1266 S O12 B K1855). By rotating the prism, different EA can be chosen. It should be noted that the viewing rectangle changes somewhat as the prisms are rotated, but, as we are using the spectrometer largely as a limb scanner, this change is not significant. The scanner has a precision of better than 0.01°.

Because the pointing accuracy and the viewing geometry are important parameters for the vertical profile retrievals (Bruns et al., 2004; Weidner et al., 2005; Baidair et al., 2013), we spend considerable effort on characterizing these parameters after installation on the aircraft. In short, an optical system consisting of a Xe-arc lamp is focused with an f = 100 mm lens on a ~0.5 mm pin-hole and then collimated via another lens (f = 100 mm) into a narrow parallel light beam with an initial diameter of 25 mm. This assembly is then placed ~15 m in front of the telescope and carefully aimed into each telescope. The absolute

angle is determined by feeding the beam through two 0.5 mm pinholes, 1 m apart, and a high accuracy inclinometer. The accuracy of this determination is ~0.15°. Each scanner has a limit switch, which allows the absolute determination of the scanning angle relative to the aircraft. During the alignment procedure, the pitch of the aircraft is thus extracted from the GH navigation system and an absolute calibration relative to the aircraft limb with an accuracy of better than 0.2° is achieved. This accuracy has been confirmed through multiple tests during the various ATTREX deployments and in the laboratory. The

elevation opening viewing angle is also accurately determined by scanning the telescopes over the light beam, while recording the light spectral radiance at the same time.

To maintain the viewing direction during flight maneuvers, an active control of the EA is used. Based on 1 Hz pitch measurements from the GH inertial navigation system, the aim of the telescopes is held to within ±0.2°, which is within the uncertainty of the elevation calibration. It should be noted that the GH flights are mostly performed with long straight flight legs and that





turns are not flown frequently. For the majority of the flights, the impact of aircraft roll does not impact our correction, and we exclude turns from the data analysis.

A special feature of the telescope assembly are diffuser plates (15 mm diameter, 1.6 mm thickness) mounted in zenith direction above each telescope that allow the measurement of direct sunlight. For the UV wavelength range, a UV grade
fused silica 220 grit-blasted optical diffuser is used. The VIS and near-IR diffusers are 220 grit-blasted soda lime float glass. The diffusers are only used when they are directly illuminated by the sun, i.e. not shaded by the aircraft fuselage. To avoid diffuse reflections from the telescope fairing, the side facing the diffusers is painted black matte. As the measurement of solar references is crucial, low solar zenith elevation direct sun measurements are performed at least once during every flight.

The three telescope/scanners protrude ~10 cm off the starboard side of the NASA Global Hawk fuselage. The telescopes are
pointed 2° away from the flight direction to avoid collecting light reflected by the fuselage. As ambient temperatures at the GH flight altitudes can drop below -80°C for extended periods of time, the telescopes are heated to above -40°C to keep the motor mechanics from freezing.

A rugged industrial computer (Moxa V2101) is used for spectrometer data acquisition, to control telescope EA, to regulate and monitor temperatures of telescopes and instrument assembly, and to communicate with the GH and the ground. Commu-
nication with the aircraft includes synchronization with the GH main clock every 90 seconds, reading of aircraft status data, including current pitch, altitude, geographic locations, etc. at 1 Hz frequency, and broadcast of instrument status to the aircraft and the ground. While the mini-DOAS is built to perform automatically, the GH communication with the ground allows a certain amount of remote control. This capability is used to select pre-programmed EA scanning strategies for different flight behavior and for solar references. These sequences are discussed in more detail below. Measured absorption spectra are saved
on a memory card in the instrument, as well as transmitted to an FTP server on-board the GH.

## 2.4   Measurement Strategy

The mini-DOAS custom software is set up to treat each spectrometer-telescope combination as an independent unit, i.e. the measurements between the different wavelength channels are not coordinated. Two main measurement strategies are used.
During level flights, the telescopes are scanned in elevation. In the case of the UV and visible, the following angles are scanned sequentially, where positive EA look upwards from the limb, while negative angles are downward viewing directions: zenith/direct sun (when suitable), 1°, 0°, -0.5°, -1°, -1.5°, -2°, -2.5°, -3°, -4°, -7°, and -15°. The IR channel is scanned in the following angles: zenith/direct sun (when suitable), -0.5°, -1°, -1.5°, -2°, -2.5°, -3°, and -4°. During ascent and descent the telescopes are kept at -0.5°. It should be noted that the -0.5° is chosen over the limb to compensate for the earth curvature, and
to provide observations which are more reflective of the flight altitude. The measurement time for each angle in the UV and VIS is set to 30 sec, while 60 sec were used in the near-IR.

At least once during each flight a longer 3 min solar reference is measured. The time of this measurement is chosen according to the flight plan, low solar zenith angles and a solar azimuth relative to the aircraft that decreased the chance of reflection from the fuselage.





## 3 Data Analysis

Scattered sunlight spectra acquired during the ATTREX flights are analyzed using the DOAS method (Stutz and Platt, 1996; Platt and Stutz, 2008). The retrieval minimizes the difference between the logarithm of the measured spectrum, after correction of electronic offset and dark current, and a model function using a combination of a linear and non-linear least squares fit. The

model function is a linear combination of the logarithm of the solar reference spectrum, also corrected for offset and dark current, a polynomial, a Ring spectrum, and various trace gas absorption cross sections. The result of this retrieval is a differential slant column density, DSCD = SCD – $SCD_{solar}$. In our case, i.e. using direct sunlight, $SCD_{solar}$ is the integral of the trace gas concentration between the instrument and the sun at the time of the observations of the solar reference spectrum. Because the Global Hawk flies above ~90% of the atmosphere, trace gas absorptions in the solar reference are quite small, but must

be considered nevertheless. The trace gas absorption cross sections in the fitting process are calculated from high-resolution absorption cross sections (Table 3) by a convolution with a measured atomic emission line, or a numerical representation of the emission line shape, in or near the wavelength window of interest. The convolution includes treatment of the $I_0$-effect (Platt and Stutz, 2008), i.e. the convolution is performed using spectrally resolved solar radiances (Kurucz et al., 1984; Kurucz, 2005). Also included in all retrievals is a spectrum of the Ring-effect (Grainger and Ring, 1962), which is calculated from each

measurement spectrum using the method described in (Bussemer, 1993; Platt and Stutz, 2008).

Individual trace gases are analyzed in different spectral windows to increase retrieval stability and decrease errors. It should be noted that the DOAS technique determines the trace gas DSCD retrieval error for each individual spectrum. The various analysis windows are described in the following sections. Please note that details on the analysis of the near-IR channel of the instrument is the topic of a forthcoming publication and is not discussed further here.

## 3.1 UV: BrO, $O_3$, and $O_4$

Three different spectral windows are used for the retrieval of $O_3$, BrO, and $O_4$ in the UV wavelength region (Table 4). All three intervals are different but show significant overlap, which simplifies the interpretation (Table 4). A polynomial of degree 2 is included, together with a solar reference and a Ring spectrum, in all analysis windows. Trace gas reference spectra are calculated using the dispersion described above and a Gaussian line-shape describing the Hg line at 366 nm (see Table 2 for

details). The use of a Gaussian line-shape, rather than the line itself, is necessary due to a slight overlap with a neighboring emission line. Inaccuracies in wavelength position due to small changes in the instrument during the observations, and errors in the wavelength calibration of the reference spectra, are corrected during the spectral retrieval procedure. A common spectral shift is used for all trace gases and a separate common spectral shift is used for the solar reference and the Ring spectrum. Typical shifts for both groups of spectra are well below 1 detector pixel.

The BrO analysis window covers 346 – 360 nm, following Aliwell et al. (2002). References of $O_3$ at 203 K and 213 K, orthogonalized to the 203 K reference, $NO_2$, and $O_4$, are included in the fit. Figure 2 shows an example for the retrieval of BrO at an EA = -0.5° during Science Flight 3 on Feb. 14, 2013 (furtheron referred to SF3-2013). BrO absorptions are small, DSCD





$= (1.94 \pm 0.19) \times 10^{14}$ molec/cm$^2$, but are clearly identified above the residual structure, which has a RMS of $3.3 \times 10^{-4}$. The average error for BrO DSCD is $2.1 \times 10^{13}$ molec/cm$^2$, with individual errors as low as $1.6 \times 10^{13}$ molec/cm$^2$.

The retrieval window for ozone has the same lower wavelength limit as that of BrO. However, a smaller upper limit is selected to improve the retrieval, as ozone only weakly absorbs above 355 nm (Table 4). While the spectral retrieval could be

improved by using the stronger ozone absorptions at lower wavelengths, the overlap is crucial for the interpretation of the BrO observations. O$_3$, BrO, NO$_2$, and O$_4$ references are included in the retrieval (Table 3). The average error of the O$_3$ DSCD in the UV is $6.4 \times 10^{16}$ molec/cm$^2$.

O$_4$ is retrieved in a wider spectral window, 338.8 – 366.3 nm, in order to allow fitting of the 340 nm and 360 nm bands together (Table 4). Due to the use of a low polynomial degree of 2, the spectral range between 347.4 – 352 nm is excluded,

which shows an unidentified residual structure that cannot be compensated with this low polynomial degree. The use of a higher degree polynomial decreases this residual structure, but causes the O$_4$ retrieval to be less stable.

## 3.2   VIS: NO$_2$ O$_3$, and O$_4$

The main focus of the observations in the visible wavelength range is NO$_2$, which impacts the partitioning of bromine species and plays an important role in ozone chemistry. Ozone and O$_4$ are also retrieved to help in the interpretation of the NO$_2$ DSCD.

This wavelength range also allows the retrieval of IO and water vapor, which is not discussed further in this manuscript. The 450.362 nm Kr lamp emission line represents the instrument function for the convolution of the high resolution reference absorption cross sections. Each trace gas is analyzed in its own wavelength window in order to optimize the spectral analysis. A polynomial of degree 2, the solar reference and a Ring spectrum and trace gas reference spectra are included in all fit windows (Table 4).

NO$_2$ has very small mixing ratios in the UT and TTL. Consequently, a low detection limit, as well as stable fit behavior near and below the detection limit, is crucial. NO$_2$ is thus analyzed in a relatively wide spectral window of 424.2 – 460.4 nm to achieve the lowest possible DSCD errors, while maintaining stability of the least squares fit. Reference spectra of NO$_2$ and O$_3$ at 203 K and 223 K (orthogonalized to the 203 K spectrum), O$_4$ and water vapor are included in the retrieval (Table 4). Figure 3 shows an example of the retrieval of NO$_2$ during SF3-2013 with a DSCD of $(5.38 \pm 0.22) \times 10^{15}$ molec/cm$^2$. The average

error for NO$_2$ DSCD is $2.6 \times 10^{14}$ molec/cm$^2$, with best detection limits as low as $2 \times 10^{14}$ molec/cm$^2$.

Ozone absorption in the visible wavelength range is analyzed in a wavelength range shifted and somewhat larger than the range for NO$_2$. The spectral range includes the larger O$_3$ absorptions between 460 – 485 nm. Cross sections of NO$_2$ at 203 K and 223 K (orthogonalized to the 203 K spectrum) are included in the fit, together with O$_4$ and water vapor (Table 4). The average error for O$_3$ DSCD is $6.2 \times 10^{17}$ molec/cm$^2$, with best detection limits as low as $2.5 \times 10^{17}$ molec/cm$^2$.

O$_4$ is analyzed with the same combination of reference spectra as NO$_2$ (Table 4) in a wavelength range of 459.8 – 488.4 nm around the 477.3 nm O$_4$ band. The average detection limit for O$_4$ DSCD is $5.6 \times 10^{41}$ molec$^2$/cm$^5$, with best detection limits as low as $4 \times 10^{41}$ molec$^2$/cm$^5$.



### 3.3 Results and Errors of DOAS retrievals

Figure 4 shows the results of the DSCD retrieval for a 2 hour section of SF3-2013 at an altitude of 17 km. The gap from 21:22 to 21:37 UT is due to the measurement of solar reference spectra, which are not shown here. During this part of the flight, the mini-DOAS was operated in the elevation scanning mode, which allows visualizing the precision of the instrument

during flight. As expected, the spectral radiance of the observations, here displayed for the visible wavelength range (Figure 4b), is lowest at the $+1°$ viewing direction and increases as the telescope looks further down into the atmosphere below the aircraft. The spectral radiance variations for EA $=-7°$ and -15 ° are caused by low clouds in the atmosphere. The $O_4$ DSCD from the visible wavelength range show a very smooth dependence on EA, which is not surprising as the statistical error of this measurement is less than 2% of the observed $O_4$ signal, even at the upward viewing angle. All variation of the $O_4$ DSCD are

thus predominately due to changes in atmospheric RT, including the presence of clouds. It should be noted that the use of the direct solar reference is highly advantageous for the detection of $O_4$, as the $O_4$ SCD of the reference is very small (less than 1 - 2 % of the total signal at the flight altitude of the Global Hawk). Both $O_3$ and $NO_2$ show the behavior expected for a trace gas with a high mixing ratio in the stratosphere, with the largest DSCD between EA = $+1°$ and -0.5°. The observations before 21:30 UT show a fairly regular dependence on EA, with superimposed random variation in the DSCD, which reflect small

variations in the atmospheric trace gas levels, as well as the statistical error in the spectral retrievals of $6.2 \times 10^{17}$ molec/cm$^2$ and $2.6 \times 10^{14}$ molec/cm$^2$, respectively. The period after 21:30 UT shows a different dependence of the $O_3$ and $NO_2$ DSCD on the EA, with a somewhat lower DSCD for the limb and -0.5° angles. We attribute this change to a transition from the lower stratosphere to the UT, which is accompanied by lower flight level mixing ratios of the trace gases. This is also confirmed by the in-situ $O_3$ observations (not shown here).

The BrO DSCD generally follows the behavior of $O_3$. However, the much higher statistical error of the observations of $2.1 \times 10^{13}$ molec/cm$^2$ make the dependence less visible. Before 21:30 UT the dependence of BrO, which has higher mixing ratios in the stratosphere than in the troposphere, on EA is as expected, i.e. higher values in the limb and at $+1°$ viewing direction. The weaker dependence after 21:30 UT is due to the generally smaller BrO concentrations, and thus a higher statistical variation.

While individual errors are not included in Figure 4 to allow a better visualization of the DSCD variation, the DOAS retrieval determines a statistical uncertainty for every spectrum (Platt and Stutz, 2008). These statistical errors will be propagated through all error calculations presented in the rest of this manuscript and in companion paper by Werner et al. (2016). Besides these errors, several other factors determine the accuracy of the DSCD observations. The accuracy of the absorption cross sections is listed in Table 3. Instrumental errors, such as spectrometer straylight, errors in correcting dark current and electronic

offset, etc., of DOAS observations are typically in the range of $2 - 3$ % (Platt and Stutz, 2008). While these effects are often difficult to quantify, the mini-DOAS is optimized to reduce these errors, for example by using filters to reduce spectrometer straylight, holding detector saturation levels constant during the measurement, and regular offset and dark current corrections. We thus estimate an upper limit of 3% for the impact of these effects on accuracy. Other uncertainties, such as the pointing accuracy, are considered in the discussion of the concentration retrievals in Section 4.



As we used direct sun spectra as solar references, it is worth discussing the impact of this approach on the uncertainty of the observations. The main motivation for using a diffuser to measure a solar reference is the higher signal to noise ratio of the solar reference, compared to a spectrum measured in the zenith for a similar exposure time. The use of a diffuser spectrum does not significantly degrade the spectral retrieval. A comparison of a spectral analysis with the solar reference from the diffuser and that using a 1° reference spectrum yields similar DSCD errors. However, the DSCD values derived using EA = + 1°spectra are generally lower than for lower EA. While this reduction is not as pronounced for $O_3$ and $NO_2$, it is more serious for $BrO$, where the change of the DSCD with EA is only half of the overall signal. The relative error of the $BrO$ DSCD is thus nearly a factor 2 smaller when the spectra are analyzed with a direct sun reference spectra.

The smaller reference SCD for a direct sunlight solar reference also offers major advantages in the trace gas concentration retrievals, as the uncertainty of the reference SCD contributes to the overall error in the concentrations. In general, a smaller reference SCD reduces the contribution of this uncertainty to the overall uncertainty of the trace gas concentrations. In addition, the RT is much simpler for a direct sun spectrum, reducing this additional uncertainty in the interpretation as well (see Section 4).

### 3.4  Additional Measurements

The retrieval of trace gas concentrations from the mini-DOAS is aided by the use of additional observations on board the GH. This section briefly describes the observations used in the rest of this study. For a more complete overview of the ATTREX GH payload, the reader is referred to Jensen et al. (2015).

Ozone concentrations in this study are measured by NOAA using a dual-beam, UV photometer (Gao et al., 2012) at sampling rates of 2 Hz at ambient pressures below 200 hPa, 1 Hz between 200 and 500 hPa, and 0.5 Hz above $\geq$ 500 hPa. The instrument has a high accuracy of 3% (except in the 300 - 450 hPa range, where the accuracy may be degraded to about 5%), and a precision of ($1.1 \times 10^{10}$ $O_3$ molecules/cm$^3$ at 2 Hz. At the typical GH flight altitude (approx. 200 K and 100 hPa) this corresponds to a precision of approximately 3.0 ppb. In-flight and laboratory inter-comparisons with existing $O_3$ instruments show that measurement accuracy was maintained in flight.

Two different instruments provide atmospheric methane ($CH_4$) observations during ATTREX. The Unmanned aircraft system Chromatograph for Atmospheric Trace Species (UCATS) measures $CH_4$ and other trace gases by gas chromatography once every 140 seconds (see details in Moore et al. (2003); Elkins et al. (1996)). In short, separation of $H_2$, CO, and $CH_4$ in air is accomplished with a pre-column of Unibeads (2 m x 2 mm diameter), and a main column of Molecular Sieve 5A (0.7 m x 2.2 mm diameter) at $\sim 110\,°C$ (Moore et al., 2003). 100 ppm of nitrous oxide is added to the ECD make-up line to improve sensitivity (Elkins et al. (1996); Moore et al. (2003)). UCATS was calibrated in-flight with a secondary $CH_4$ standard after every three ambient air measurements. These calibrations are used to correct instrumental drift. Precision of the UCATS $CH_4$ observations is $\pm 0.5\%$. Measurements are traceable to the WMO Central Calibration Laboratory (CCL), and are reported on the $CH_4$ WMO X2004A scale (Dlugokencky et al., 2005) (with update given at http://www.esrl.noaa.gov/gmd/ccl/ch4_scale.html).

Measurements of $CH_4$ are also performed by Harvard University using a pressure and temperature stabilized Picarro Cavity Ringdown Spectrometer (G2401-m, Picarro Inc., Santa Clara, CA, USA). The HUPCRS instrument uses Wavelength-Scanned



Cavity Ringdown Spectroscopy (WS-CRDS) technology to make high precision measurements (Crosson (2008), Rella et al. (2013), Chen et al. (2013)) of $CO_2$, $CH_4$, and CO concentrations every $\sim 2.2$ seconds. The data are reported as 10 second averages. In-flight precision for $CH_4$ is 0.2 ppb.

### 3.5 Modeling Tools

5    In order to convert the observed trace gas DSCD into concentrations, various retrieval methods are applied which are based on a RTM. The RTM is capable of quantitatively describing the observed mini-DOAS radiances and trace gas absorptions. As far as possible, this model is constrained by observations. However, because the mini-DOAS observations are also sensitive to trace gas absorptions above and below the GH, we also employ a 3D atmospheric chemistry model to vertical trace gas concentration profiles. Both model frameworks are described in the following sections.

### 10    3.6 Radiative Transfer Model

The received limb radiances are modeled in 1D and, in selected cases in 3D, using version 3.5 of the Monte Carlo radiative transfer model McArtim (Deutschmann et al., 2011). The model's input is chosen according to the on-board measured atmospheric temperatures and pressures, including climatological low latitude aerosol profiles from SAGE III (http://sage.nasa.gov/missions/about-sage-iii-meteor-3m/), and lower atmospheric cloud covers (mostly marine stratocumulus (mSc) clouds), as indicated by the cloud physics lidar measurements made from aboard the GH (see https://eosweb.larc.nasa.gov/project/sage3/sage3_table). The RTM is further fed with the actual geolocation of the GH, solar zenith and azimuth angles as encountered during each measurement, the telescopes azimuth and EA, as well as the field of view (FOV) of the mini-DOAS telescopes. In the standard run, the ground (oceanic) albedo is set to 0.07 in the UV, and 0.2 in the VIS. For the simulations of the BrO, $O_3$, and $NO_2$ absorptions, the RTM is further fed with TOMCAT/SLIMCAT simulated vertical concentration profiles of the targeted gases simulated along the GH flight paths. Figure 5 visualizes the RT for GH limb measurements at 18 km altitude based on a detailed 3D RT simulation. The simulation demonstrates that the Earth's sphericity, the correct treatment of atmospheric refraction, cloud cover, ground albedo, etc. are relevant in the context of the UV/VIS/near-IR limb measurements at the GH flight altitude (Deutschmann et al., 2011). Past comparison of measured and McArtim modeled relative limb radiances have confirmed the quality of the RT simulations with McArtim (e.g., see Fig. 5 and Fig. 6 in Deutschmann et al. (2011) and Fig. 2 25 in Kreycy et al. (2013)).

### 3.7 Photochemical modeling

Simulations of the TOMCAT/SLIMCAT 3-D chemical transport model (CTM) are used to provide vertical trace gas profiles for the RTM calculations (Chipperfield, 1999, 2006; Chipperfield et al., 2015). We are particularly interested in the simulations of $O_3$, BrO, and $NO_2$, but results of $CH_4$ are also used to constrain small variations in meteorology. In the runs used here 30   TOMCAT/SLIMCAT model meteorology, i.e. large-scale winds, temperatures, as well as convective mass fluxes, is driven by ECMWF ERA-interim reanalyses. Detailed stratospheric chemistry is included in the model, based on the JPL-2011 kinetic





and photochemical data, (Sander et al., 2011) including recent updates. The model chemistry is constrained by prescribed time-dependent surface mixing ratios. The following values are assumed for brominated organic species: $[CH_3Br]$ = 6.9 ppt, [halons]= 7.99 ppt, $[CHBr_3]$ = 1.0 ppt, $[CH_2Br_2]$ = 1.0 ppt, and $[CHClBr_2, CHCl_2Br, CH_2ClBr, ....]$, which together contain 1 ppt of bromine atoms, in agreement with recent reports (e.g., WMO (2015), Sala et al. (2014)). The sum of all organic bromine

at the surface is $[Br_y^{org}]$ = 20.89 ppt . We include 0.5 ppt BrO in the troposphere, in agreement with the finding discussed below (section 4.4). Other sources of bromine for UT, LS, and TTL are not included in the model runs (e.g., Fitzenberger et al. (2000), Salawitch et al. (2010), Wang et al. (2015), and others). Global mean $CH_4$ surface concentrations are specified based on AGAGE (https://agage.mit.edu/) and NOAA observations, and include recent $CH_4$ growth rate variations. The model run used for the trace gas retrievals (#583) is initialized in 1979 with a spin-up time of 34 years at low horizontal resolution ($5.6°$

$\times 5.6°$), and with 36 unevenly spaced levels in the altitude range $0 - 63$ km. Model output for January 1, 2013 was interpolated to a horizontal grid of $1.2° \times 1.2°$ and the model run was continued at this higher resolution through the end of the ATTREX campaign. The modeled vertical trace gas profiles for the measurement time and location of the mini-DOAS observations are stored as input for the RTM. Figure 6 shows such a dataset for SF3-2013. It should be noted that the model output confirms our initial qualitative interpretation of the raw data in Figure 4, that the GH transitioned from the lower stratosphere to the UTLS

region around 21:30 UT.

## 4   Concentration retrievals

Calculating concentrations and concentration profiles from the differential slant column densities, DSCD, derived in the DOAS retrievals is a multi-step process (Platt and Stutz, 2008). The first step is to determine the slant column density of the solar reference in order to determine the total slant column density (SCD) of a trace gas at the time of observation (Eq. 1). Because

the mini-DOAS uses direct sunlight for the solar reference observation, the $SCD_{ref}$ are calculated from the simulated TOM-CAT/SLIMCAT profiles of the overhead $O_3$, $NO_2$, and BrO concentrations at the time and location the measurement (Figure 6), using the direct optical path between the instrument and the sun. The $O_3$, $NO_2$, and BrO $SCD_{ref}$ for SF3, for which the DSCD are shown in Figure 4, are $7.6 \times 10^{18}$ molec/cm$^2$, $3.0 \times 10^{15}$ molec/cm$^2$, and $2.2 \times 10^{13}$ molec/cm$^2$, respectively. The $SCD_{ref}$ are of similar magnitude for other flights.

A comparison with the DSCD in Figure 4 shows that the $SCD_{ref}$ are generally smaller than the DSCD around the limb, while they are of similar size as the $O_3$, $NO_2$, and BrO DSCD at lower EA. Because the $SCD_{ref}$ is determined for a direct solar observation, its uncertainty is solely determined by the model uncertainty. The $SCD_{ref}$ error, as well as its impact onto the overall error of the retrieved trace gas concentrations, is discussed in Section 4.4.

Two measurement strategies are adopted depending on flight mode (Section 2.4): Elevation scans during level flight, and

limb observations during aircraft ascent/descent. Section 4.1 discusses the challenges encountered in interpreting the SCD of elevation scans during ATTREX. Trace gas profile retrievals from limb SCD during aircraft ascent and descent based on optimal estimation inversions are discussed in Section 4.2. Finally, a new approach to derive trace gas concentrations using an



$O_3$-scaling technique of the limb observations is presented along with a discussion of the uncertainties and advantages of this novel approach (Section 4.3).

## 4.1 Optimal Estimation Retrievals from Elevation Scans

The idea of using elevation scans of a DOAS instrument to obtain information on vertical trace gas distributions is well established for balloon-, and ground-based instruments (Wagner et al., 2004; Weidner et al., 2005; Platt and Stutz, 2008; Kritten et al., 2010; Kreycy et al., 2013) and accordingly has also been proposed for aircraft observations (Bruns et al., 2004, 2006). For trace gases with high concentrations in the stratosphere, i.e. $O_3$, $NO_2$, and $BrO$, the highest DSCD are found for the largest viewing angles and a clear reduction is seen for downward viewing direction (Figure 4). The opposite behavior is found for $O_4$ and the spectral radiance. For the interpretation of this information with respect to concentration profiles, a two step retrieval is implemented. The first step aims to determine the RT conditions of a specific scan using a known tracer, which in our case is $O_4$ or the relative spectral radiances. Once the RT is constrained with sufficient accuracy, the trace gas concentration profile can be retrieved in the second step.

We first assess the ability of our observations to constrain the RT by using $O_4$ through theoretical calculations. Figure 7a shows the contribution of the observed $O_4$ optical density from various altitudes, i.e. the altitude specific sensitivity multiplied with the product of the $O_4$ concentration and the absorption cross section, for a cloud free location above the ocean. The sensitivity to different altitudes, in particular in the range between $10 - 17$ km can clearly be seen. However, it is also obvious that a very large portion of the $O_4$ absorption originates from below 10 km. For angles larger than -4° the contribution above and below 10 km is similar. This, however, changes drastically when clouds are present in the lower atmosphere (Figure 7 b to f). As the optical thickness of lower atmospheric clouds increases, the contribution of the lower atmosphere becomes more and more prevalent, to the point where the lower atmosphere dominates the $O_4$ signal. The same behavior can be observed in the relative spectral radiances (not shown). The $O_4$ and the spectral radiance are thus highly sensitive to the RT in the lower atmosphere, and in particular to the presence of clouds. This can also be directly seen in the observational data (Figure 4). The elevation scans of $O_4$ and the spectral radiance vary greatly in this two hour period, due to changes in cloud cover below the aircraft.

The main challenge in the interpretation of the DSCD from the elevation scans is thus to accurately describe the RT along the line of sight of the mini-DOAS, which can be several hundred kilometers in length, during an elevation scan. This problem is common for limb observation geometries, see for example (Oikarinen, 2002). In an ideal situation, i.e. a cloud free atmosphere over the length of the line of sight, this retrieval can be performed using $O_4$ or relative radiances. However, we did not encounter such ideal clear sky conditions during ATTREX. The use of $O_4$ DSCD or relative radiances to constrain the RT thus introduces considerable uncertainties due to the poorly constrained and inhomogeneous cloud cover below the aircraft. This problem is magnified by the fact that the cloud cover in the line of sight will change as the aircraft moves, thus introducing variation within one elevation scan that cannot be corrected and/or distinguished from changes in other parameters.

We thus conclude that, while the elevation scans clearly contain information on the altitude distribution of the trace gases of interest, the uncertainty of the RT calculations is too large to allow the retrieval of $BrO$ and $NO_2$ concentrations at the accuracy





required for our study. It should be noted that the problem with $O_4$, which is typically the tracer of choice to constrain the RT in most DOAS applications, is its highly altitude dependent profile, i.e. highest levels at the surface and a rapidly decreasing concentration with altitude. This makes it unsuitable for high-altitude aircraft DOAS applications.

### 4.2 Optimal Estimation Retrievals during Ascent/Descent

While the decrease of retrieval quality due to low level clouds is fairly obvious for elevation scans, this issue is more difficult to assess for vertical profile retrievals from limb observations during ascent and descent of the aircraft. Previous publications have used this approach and found it to be suitable only under completely cloud-free conditions (Volkamer et al., 2015). However, as mentioned earlier, we did not encounter such conditions on spatial scales equivalent to our viewing geometry, i.e. no cloud free conditions along the 200 - 300 km viewing length, during ATTREX. We therefore implement a full profile retrieval algorithm for the ATTREX ascent and descent data. This algorithm follows the two step approach for (1) atmospheric extinction and (2) trace gas retrieval outlined earlier. It is based on the optimal estimation technique (Rodgers, 2000), which seeks to minimize the square of the difference of measured quantities $\mathbf{y}$, including Gaussian distributed errors $\epsilon$ (i.e., the measured SCD with their errors) with a forward model function $\mathbf{F}(\mathbf{x}, \mathbf{b})$.

$$\mathbf{y} = \mathbf{F}(\mathbf{x}, \mathbf{b}) + \epsilon \tag{2}$$

For the first step of the retrieval the targeted quantity $\mathbf{x}$ to be optimized is the aerosol extinction profile, and the fixed model parameters, $\mathbf{b}$, are atmospheric conditions, i.e. profiles of temperature and pressure, albedo, etc. The measurement vector $\mathbf{y}$ of dimension m are the observed DSCD of $O_4$, at a wavelength close to an absorption band of the trace gases targeted in Step 2, during an ascent or descent (Greenblatt et al. (1990), Pfeilsticker et al. (2001), Thalman and Volkamer (2013). The measurement errors ($\epsilon$) are the error of the $O_4$ DSCD. In this case $\mathbf{F}(\mathbf{x}, \mathbf{b})$ is a non-linear function of $\mathbf{x}$, a non-linear retrieval, most often using a Levenberg-Marquardt iteration scheme, is employed. The numerical iteration starts with an a-priori vector $\mathbf{x}_1 = \mathbf{x_a}$

$$\mathbf{x}_{i+1} = \mathbf{x}_i - \left[ (1+\gamma)\mathbf{S}_a^{-1} + \mathbf{K}_i^T \mathbf{S}_\epsilon^{-1} \mathbf{K}_i)^{-1} \right] \cdot \left\{ \mathbf{K}_i^T \mathbf{S}_\epsilon^{-1} (\mathbf{F}(\mathbf{x}_i) - \mathbf{y}) - \mathbf{S}_a^{-1} (\mathbf{x}_i - \mathbf{x}_a) \right\} \tag{3}$$

The parameter $\gamma$ weights the strength of the a priori covariance used in the gradient method and the Gauss-Newton part of the Levenberg-Marquardt iteration to improve convergence of the solution. The solution of this iteration is an vertical atmospheric extinction profile and its uncertainty. In the second step of the retrieval, i.e. the retrieval of trace gas profiles, $\mathbf{x}$ is the trace gas concentration profile, while $\mathbf{b}$ again describes the atmospheric conditions, but now includes the results from Step 1, i.e. the extinction profile. The measurement vector $\mathbf{y}$ and errors $\epsilon$ are the observed DSCD and errors of the respective trace gas. In this case $\mathbf{F}(\mathbf{x}, \mathbf{b})$ is a linear function of $\mathbf{x}$ and a linear solution can be used:

$$\hat{\mathbf{x}} = (\mathbf{S}_a^{-1} + \mathbf{K}^T \mathbf{S}_\epsilon^{-1} \mathbf{K})^{-1} (\mathbf{K}^T \mathbf{S}_\epsilon^{-1} \mathbf{y} + \mathbf{S}_a^{-1} \mathbf{x}_a) \tag{4}$$

In both equations, $\mathbf{K}$ is the Kernel matrix ($K_{ij} = \partial F_i(\mathbf{x})/\partial x_j$), and $\mathbf{S}_\epsilon$ is the measurements covariance matrix. In cases where the problem is ill-posed (i.e., n, the dimenion of $\mathbf{x}$, is larger than the rank of matrix $\mathbf{K}$), a priori estimates of the state $\mathbf{x}_a$ with covariance $\mathbf{S}_a$, which represent the best knowledge of $\mathbf{x}$ before the measurement, are included in the solution.





Our initial retrievals of various ATTREX ascents and descents using this approach revealed that, under the assumption that we can accurately determine the aerosol extinction profile in the first step, a $BrO$ concentration profile retrieval with the required accuracy is possible. However, further investigation of the uncertainty in the aerosol extinction profile retrieval due to clouds showed that the propagation of this error onto the $BrO$ concentration retrieval severely increases the $BrO$ error. In the

following section the impact of this error propagation is discussed in more detail by considering a theoretical case with a low cloud cover of optical depth (OD) of $\tau_{mSc} = 10$ at 1 km altitude, and an aerosol extinction profile from the SAGE II climatology in the tropics. The RT model is run in forward mode to calculate the $O_4$ limb DSCD the mini-DOAS would have observed during a GH ascent from 14.5 km to 17.5 km. These theoretical $O_4$ DSCD are then used in three sensitivity retrievals, where the cloud OD is held fixed at $\tau_{mSc} = 10$ and alternatively at 30% smaller and larger OD values, i.e. $\tau_{mSc} = 7$ and $\tau_{mSc} = 13$.

These two values are chosen because they reflect the typical uncertainty of an extinction retrieval in the lower atmosphere that includes optimization of low cloud OD. The two runs thus exemplify the influence of the uncertainty of the cloud OD retrieval onto the rest of the aerosol profile. Figure 8 shows these results of these two retrievals for the aerosol extinction in the upper atmosphere. The aerosol extinction between 15.5 - 16.5 km for the case with $\tau_{mSc} = 13$ is zero, compared to the results of the $\tau_{mSc} = 10$ retrieval of 0.001. In the case of the $\tau_{mSc} = 7$ retrieval, the aerosol extinction is 5 - 10 times larger than the $\tau_{mSc} = $

10 value. It is thus clear that a 30% uncertainty in the low cloud OD leads to very large uncertainties in the aerosol extinction at the GH flight altitude.

It should be noted that, while these result are based on a fairly simple idea of comparing three cases, the general conclusion that uncertainties in the determination of low cloud OD retrievals will propagate onto the aerosol extinction profile in the upper atmosphere for high-altitude limb observations applies to any source of uncertainty. These uncertainties can stem from the

general error of an non-linear extinction profile retrieval, which we determine to be around 30%, but also from changes in low cloud OD and cloud coverage during the ascent, which can be up to 30 min long and cover distances of nearly 200 km.

In order to determine how the aerosol extinction profile uncertainty propagates to $BrO$ concentration errors, a forward RT run with the $\tau_{mSc} = 10$ profile and a typical $BrO$ concentration profile is used to calculate theoretical mini-DOAS $BrO$ DSCD during the ascent. These values are then used in three sensitivity $BrO$ profile retrievals using the $\tau_{mSc} = 7, 10,$ and 13

extinction profiles from the aerosol extinction test. The $\tau_{mSc} = 10$ retrieval results agree well with the original profile, while the $BrO$ concentration in the 17 km altitude range are smaller by more than 0.5 ppt for $\tau_{mSc} = 7$ and larger by more than 0.5 ppt for $\tau_{mSc} = 13$ (Figure 8). This illustrates how the uncertainty in lower cloud OD impacts the $BrO$ retrieval. Because the two step approach does not propagate the aerosol extinction error onto the trace gas retrieval error, this uncertainty is typically not considered. It should be noted that OD variations can be larger under cloudy conditions and that the error can thus be

considerably larger. In addition, this uncertainty is only one component of the overall error, and the final mixing ratio error is thus higher. Our tests do not allow a general conclusion on this error propagation for all possible DOAS viewing geometries. However, for the GH altitude and a limb observational strategy, the uncertainty of low cloud OD, or along the same line of argument the possibility of an OD change during an ascent or descent, introduces errors in the $BrO$ concentration which are larger that those needed for the investigation of UTLS bromine chemistry.





In conclusion, the presence of low clouds introduces uncertainties in optimal estimation trace gas retrievals based on observed $O_4$ DSCD that are difficult to overcome. As illustrated above, the cloud OD needs to be better known than can currently be determined to achieve $BrO$ mixing ratio errors of 0.5 ppt or below. Much of this effect stems from the strongly altitude dependent $O_4$ concentration profile (e.g., Pfeilsticker et al. (2001)). The presence of low clouds leads to an increase of the

lower atmosphere albedo (Los and Duynkerke, 2001), as well as a signal from multiple scattering in the cloud, both of which increase the weight on the lower atmospheric portion of the observed $O_4$ DSCD. This change is counteracted in the retrieval by increasing/decreasing the upper atmospheric aerosol scattering, thus changing the RT in the limb geometry. It is thus necessary to overcome this problem and to use a different retrieval approach, for example based on a different trace gas. We thus develop a novel and more robust approach for the concentration retrieval based on the relation of DOAS $O_3$ DSCD and in-situ

observations on the GH.

### 4.3   The $O_3$-Scaling Method

In order to reduce the trace gas retrieval uncertainties for the high-altitude GH observations, the so-called $O_3$-scaling technique is introduced (e.g., Raecke (2013), Großmann (2014), and Werner (2015)). This method makes use of the in-situ $O_3$ measured by the NOAA instrument (Section 3.4) and the limb $O_3$ slant column densities, $SCD_{O3}$, simultaneously measured in the UV

and visible wavelength range. The basic idea behind this method is that the ratio $SCD_{O3}/[O_3]$ is a proxy for the effective (horizontal) light path length at flight altitude for a given wavelength. This effective path length can then be used to derive $BrO$ and $NO_2$ concentrations from the SCD retrieved in the same wavelength interval as ozone. In reality the situation is more complicated, as the limb observations are also sensitive to the absorption in air above and below the aircraft. However, because of the extremely long light paths in the limb, or in our case at -0.5°, the $SCD_{O3}$ is highly dependent on flight altitude

$[O_3]$. This is illustrated in Figure 9, which compares the ozone SCD to in-situ $[O_3]$ during an ascent/descent maneuver during SF3-2013, when $SCD_{O3}$ is highly correlated with $[O_3]$. This supports the idea that $SCD_{O3}$ is predominately sensitive to $O_3$ at flight altitude, and that the ratio of the two parameters are a measure of RT conditions.

The advantage of the $O_3$-scaling over the optimal estimation method comes from reducing potential uncertainties in the RT (c.f., due to aerosols and cloud particles) which affect the measurements of $O_3$ and the gases $BrO$ and $NO_2$ in the same

wavelength interval equally. This is particularly important for situations with horizontal heterogeneities of the RT conditions, such as broken cloud cover, which cannot be described accurately by 1-D altitude based optimal estimation retrievals. In addition, changing RT conditions of the atmosphere during a single observation period (ascent or descent of the aircraft) yields ambiguous and ill-constrained results for an optimal estimation retrieval (see above). This shortcoming is avoided by the $O_3$-scaling technique because a trace gas concentration is retrieved for each single measurement. In this section the theoretical

basis of this method is described, its application on the ATTREX 2013 observations, and provide a careful analysis of the uncertainties.

Due the wavelength dependence of Rayleigh and Mie scattering, the $O_3$-scaling technique is most accurate when the $O_3$ SCD are retrieved in similar wavelength intervals (UV: 346 - 355 nm, VIS: 437 - 485 nm) as the target trace gases e.g., $BrO$ in the UV (343 - 355 nm), and $NO_2$ in the visible wavelength range (424 - 460 nm) (see Section 3). While the spectral overlap



between $O_3$ and $BrO$ is very good, there is a slight offset between $O_3$ and $NO_2$ in the visible wavelength range, as the strong $O_3$ and $NO_2$ absorptions are close, but do not overlap.

The mathematical formalism on which the $O_3$-scaling technique is based upon, is thus developed for two different trace gases absorbing at either the same wavelengths (350 nm in the UV) or two similar wavelengths in the VIS ($O_3$ 461 nm, $NO_2$:

436 nm). The SCD (= dSCD + $SCD_{ref}$) of a specific trace gas is the sum of slant column amounts ($[X]_i \cdot B_{X_i} \cdot z_i$) of individual atmospheric layers $i$ of thickness $z_i$, concentrations $[X]_i$, and so-called box air mass factor $B_{X_i}$ for the targeted gas X (here $BrO$ and $NO_2$) and the scaling gas P (here $O_3$) over the entire height of the atmosphere:

$$SCD_X = \sum_i [X]_i \cdot B_{X_i} \cdot z_i \tag{5}$$

$$SCD_P = \sum_i [P]_i \cdot B_{P_i} \cdot z_i, \tag{6}$$

The box air mass factor, $B_{X_i}$, is the ratio of the effective slant absorption path in the atmospheric layer i and the layer thickness (Platt and Stutz, 2008). It is also a measure of the sensitivity of the observation to a specific layer, with the layer at flight altitude having, by far, the longest slant absorption path and the largest B for the used EA = -0.5 °. For the flight altitude layer, $j$, the concentrations for both gases can be expressed as

$$[X]_j = \frac{SCD_X - \sum_{i \neq j}[X]_i \cdot B_{X_i} \cdot z_i}{B_{X_j} \cdot z_j} \tag{7}$$

$$[P]_j = \frac{SCD_P - \sum_{i \neq j}[P]_i \cdot B_{P_i} \cdot z_i}{B_{P_j} \cdot z_j}. \tag{8}$$

For weak absorbers (i.e. those with optical densities much smaller than unity), the box air mass factors $B_{X_j}$ and $B_{P_j}$ are equal if both gases X and P are retrieved in the same wavelength interval. In the case of a small difference in the retrieval wavelength range, $B_{X_j}$ and $B_{P_j}$ are very similar, for example in our VIS case the difference is $2 - 5\%$ above 16 km altitude and $\sim 7\%$ at 14.5 km according to RT calculations for the atmospheric conditions during ATTREX. One can thus apply the approximation

that $B_{X_j}$ and $B_{P_j}$ are equal within the uncertainty of the retrieval. However, it should be noted that this approximation must be checked for non-overlapping wavelength ranges before applying the $O_3$-scaling technique. The ratio of $[X]_j$ and $[P]_j$ can then be simplified to:

$$\frac{[X]_j}{[P]_j} = \left( \frac{SCD_X - \sum_{i \neq j}[X]_i \cdot B_{X_i} \cdot z_i}{SCD_P - \sum_{i \neq j}[P]_i \cdot B_{P_i} \cdot z_i} \right) \tag{9}$$

Further, by defining so-called $\alpha$-factors ($\alpha_X$, and $\alpha_P$), which describe the fraction of the absorption in layer $j$ of the total

atmospheric absorption for both gases

$$\alpha_{X_j} = \frac{SCD_X - \sum_{i \neq j}[X]_i \cdot B_{X_i} \cdot z_i}{SCD_X} \tag{10}$$

$$= \frac{[X]_j \cdot B_{X_j} \cdot z_j}{\sum_i [X]_i \cdot B_{X_i} \cdot z_i} \tag{11}$$





and

$$\alpha_{P_j} = \frac{SCD_P - \sum_{i \neq j}[P]_i \cdot B_{P_i} \cdot z_i}{SCD_P} \tag{12}$$

$$= \frac{[P]_j \cdot B_{P_j} \cdot z_j}{\sum_i [P]_i \cdot B_{P_i} \cdot z_i}. \tag{13}$$

the master equation of the $O_3$-scaling technique can be simplified to

$$[X]_j = \frac{\alpha_{X_j}}{\alpha_{P_j}} \cdot \frac{SCD_X}{SCD_P} \cdot [P]_j \tag{14}$$

In our case, $SCD_X$ and $SCD_P$ are obtained from the DOAS DSCD retrievals and $SCD_{ref}$ calculated from the TOMCAT/SLIMCAT model results of the vertical trace gas profile above the aircraft and the airmass factor for a direct sun observation. $\alpha_{P_j}$ and $\alpha_{X_j}$ are determined using Equations 11 and 13 using wavelength dependent RTM simulations constrained by the trace gas vertical profiles from the TOMCAT/SLIMCAT model (section 3.7).

In order to determine $[P]_j$ one needs to consider that the mini-DOAS is sensitive to air several hundred kilometers ahead of the aircraft, and that the measured in-situ concentration at the time of the DOAS measurement is not necessarily representative of the air mass probed by the mini-DOAS. However, during level flights one can use the in-situ ozone concentration observed after the DOAS measurement to gain this information. We thus use the RTM to derive a spatial sensitivity curve to the in-situ observations ahead of the aircraft. Figure 10 shows such a curve for the visible spectral range, expressed as time after the DOAS observation. $[P]_j$ is then calculated as the $O_3$ in-situ concentration weight-averaged by this function or, in mathematical terms, the convolution of the $O_3$-time data set with the function shown in Figure 10. In the case of aircraft ascent and descent, the line-of-sight averaging is not suitable because air masses probed by our instrument are not probed later by the in-situ instrument, and hence the in-situ $O_3$ concentrations averaged over the measurement time are used for the scaling calculation. Under most conditions this approach is suitable, as the ozone concentrations before and after the dive are similar. However, care must be taken when a change of airmass regime is encountered during the dive maneuver, as for example during Descent 1 in SF3-2013, when the GH crossed from the lower stratosphere into the TTL.

Many of the details of the $O_3$-scaling technique and Equation 14 are included in the $\alpha$-factor ratio. In general terms, $\alpha$-factors quantify the fraction of the limb SCD originating from the atmospheric layer at flight altitude. The main factors influencing $\alpha$ are the vertical trace gas profile and the RT conditions. For weak absorbers at one, or very similar wavelengths, the RT is the same for all trace gases. Consequently, the RT affected by aerosols and clouds largely cancel out in the $\frac{\alpha_{X_j}}{\alpha_{P_j}}$ ratio, leaving the vertical concentration profile as the main factor impacting the $O_3$-scaling technique.

Figure 11 displays one simulation of the $\alpha$-factors for the limb measurements of $O_3$, and BrO in the UV spectral range, and $O_3$ and $NO_2$ in the visible spectral range for the sunlit part of SF3-2013. The figure indicates the varying sensitivities of the limb measurements as a function of relative layer concentration, as compared to whole atmospheric (mostly overhead) concentration of the targeted gas and the RT conditions. $\alpha$-factors are large (0.3 to 0.6) for measurements within the extra-tropical lowermost stratosphere (from 18:00 - 23:20 UT), and smaller (0.02 to 0.3) for measurements in the TTL (23:10 - 4:30 UT with 3 ascent/descent maneuvers at 23:10 - 00:10 UT, 00:45 - 01:45 UT, and 02:30 - 4:30 UT). The low $\alpha$-values indicate that, even though the line-of-sight within the limb layer $j$ is very large (of the order of hundreds of kilometers), the





concentration of the targeted gas is small compared to other (mostly overhead) located atmospheric layers. However, all $\alpha$'s show very similar behavior, reflecting the high similarity in the RT, as well as the general vertical distribution of the trace gases. This variation is thus much reduced in the $\frac{\alpha_{X_j}}{\alpha_{P_j}}$ ratio, as the RT effects mostly cancel out. Nevertheless, limb measurements are sensitive to any uncertainties in the RT and trace gas concentrations in the atmospheric layers above and below the GH.

Consequently a detailed investigation of the statistical uncertainties is necessary (section 4.4).

One aspect of the $O_3$-scaling technique that requires addition scrutiny is the performance of the TOMCAT/SLIMCAT model relative to the observations, as the RT calculations that are used for the determination of the $SCD_{ref}$ and $\alpha$-factors depend on the modeled vertical trace gas profiles. We thus compare the TOMCAT/SLIMCAT simulations of $CH_4$ (HUPCRS) (Trace b), $O_3$ (NOAA) (Trace c) with the in-situ observation along the flight track (Figure 12). Overall the data compare reasonably well, but

clear differences in the measured and simulated gas concentrations are seen as well. The difference between the measured and modeled trace gas concentration can have three different sources: (1) deficiencies in the source gas concentrations at the surface assumed in the model, (2) deficiencies in incorrectly simulating the vertical dynamics in the model, and/or (3) deficiencies of the adopted photochemistry. In the case of $CH_4$ the global surface concentrations as well as the slow photochemical degradation are reasonably well-known. The vertical dynamics of the model and its resolution is thus identified as the most likely cause of

the discrepancies. To remedy the deficits in correctly representing the vertical, especially small-scale, dynamics the modeled $O_3$ profile are vertically shifted until measured and modeled $O_3$ agree (Figure 13). All of the other modeled profiles are vertically shifted by the same amount. Sensitivity tests using $CH_4$ as the proxy for the vertical shift lead to very similar results. However, because the $O_3$ concentrations are more strongly monotonically increasing with altitude in the probed altitude range, $O_3$ is used for the altitude adjustment. It should be noted that the vertical shift of the profiles required to match the modeled

and observed $O_3$ is mostly smaller or on the order of the vertical resolution of the model ($\sim$1 km in the UTLS), and in rare cases reach up to 1.5 km. Once the model's deficiencies in vertical dynamics are removed, measured and TOMCAT/SLIMCAT modeled $CH_4$ largely agree (Figure 13).

The impact of this sub-grid scale correction of the model results on BrO and $NO_2$ concentrations is illustrated in Figure 12 and Figure 13. In the case of BrO there is a small change in both the simulated and the retrieved mixing ratios. The change

is more obvious for model simulated $NO_2$, while the retrieved BrO does not change as much. The reason for the change in modeled $NO_2$ is the large vertical concentration gradient between the upper troposphere and lower stratosphere.

### 4.4 Sensitivities and errors in the trace gas retrieval

The error and uncertainties going into the $O_3$-scaling technique and its sensitivity towards all input parameters are analyzed based on Gaussian error propagation in Equations 1, 11, 13, and 14.

$$[\Delta X]_j = [(\frac{\Delta \alpha_{X_j}}{\alpha_{X_j}})^2 + (\frac{\Delta \alpha_{X_P}}{\alpha_{X_P}})^2 + (\frac{\Delta SCD_X}{SCD_X})^2 + (\frac{\Delta SCD_P}{SCD_P})^2 + (\frac{\Delta [P]_j}{[P]_j})^2]^{0.5} \times [X]_j \tag{15}$$

A number of different sensitivity calculations are necessary to quantify the impact of different sources of errors on the derived trace gas concentrations (see Table 5 and figures in supplement). Major errors and uncertainties are due to spectral retrieval error (see Section 3.3) which are directly propagated into the final error. However, for small $[X]_j$ the errors due to uncertainties





of $\Delta SCD_{ref}$, i.e. from the vertical trace gas profile above the aircraft, have to be considered (Runs 3 to 6, Figures 2 and 11 in the supplement). We thus vary the TOMCAT/SLIMCAT simulated overhead slant column amount within their likely errors, i.e. $\pm$ 15% for both $NO_2$ and BrO (Runs 3 and 4), as well as an extreme case of i.e. $\pm$ 30% (Runs 5 and 6). For the more realistic $\pm$ 15% case, the absolute errors introduced for $NO_2$ and BrO are in the range of $0 - 2$ ppt and $0.06 - 0.2$ ppt, respectively.

The impact for the more extreme case is approximately twice as high. This error is slightly altitude dependent, with errors at 17.5 km at the higher limit of the range and those at 14.5 km at the lower end of the range (black points in Fig. 14). Similar tests changing the overhead $O_3$ profile by $\pm$ 3% (Runs 7 and 8) yield $NO_2$ and BrO errors in the range of $0 - 2$ ppt and $0 - 0.2$ ppt, respectively. A more extreme, but less likely case of $\pm$ 10% (Runs 9 and 10) variation leads to values about twice this size. In contrast, the error in the in-situ measured ozone (Runs 1 and 2), the amount of aerosol and cirrus particles in flight altitude

(Runs 14, 15, and 19 - 21), the occurrence of marine strato-cumulus clouds (Runs 16, 17, and 18), changes in the ground albedo (Runs 22, 23, and 24), and the pointing errors of the telescopes (Runs 25 and 26) seem to play a minor role in the error budget, mostly because the uncertainties of the $\alpha$-factors cancel out in the $\alpha$ ratio used in equation 14. The largest uncertainties, aside from the DOAS retrieval error, are the tropospheric trace gas levels of $NO_2$, and BrO in the model. While this seems somewhat counter-intuitive, a certain number of upwelling photons carrying some $NO_2$, and BrO absorption from the troposphere may

contaminate the limb observations made at EA = - 0.5°. Since this quantity is ill-constrained, three sensitivity simulations are performed (Runs 11, 12, and 13) with tropospheric $NO_2$, and BrO mixing ratios of 10/15/20 ppt and 0.1/1/1.5 ppt, respectively (Figure 5 and 14 in the supplement). These runs show a considerable impact on the trace gas mixing ratios at flight altitude. The limb-scanning observations made in the low level part (at 14.2 km altitude) of the dive #2 during SF3-2013 (Figure 15) are thus used to further constrain the tropospheric mixing ratios. Here, for the measurements made at EA = 1°, -0.5°, -1.0°,

-1.5°, -2.0°, -2.5°, -3.0°, -4.0°, -7.0°, and -15.0°, the DSCD of $NO_2$ and BrO are evaluated against a limb spectrum taken at EA = 0°, and compared to simulated DSCD assuming the prediction of TOMCAT/SLIMCAT, or constant mixing ratios located lower in the troposphere (Figure 15). The comparison indicates that $NO_2$ mixing ratios below the aircraft are below 10 ppt, and BrO mixing ratio are around $0.5 \pm 0.5$ ppt, with an indication of somewhat larger BrO mixing ratios (but $\leq 1.5$ ppt) possibly being present just below the flight altitude. The contribution of the tropospheric trace gas concentration error to the mixing

ratio uncertainty at flight altitude (see red points in Figure 14), is approximately altitude independent at $\sim 12$ ppt for $NO_2$ and decreases from $\sim 0.5$ ppt at 14.5 km to $\sim 0.25$ ppt at 17.5 km for BrO.

The overall error of the retrieved trace gas concentrations is determined by Gaussian error progation of all errors, including the spectral retrieval error, as shown in Figure 13 as a gray bar. The typical error for flight level $NO_2$ and BrO are $\pm 15$ ppt and $\pm 0.5$ ppt, respectively.

## 5   Results

While it goes beyond the scope of this manuscript, we briefly discuss the results of the BrO and $NO_2$ retrievals from SF3-2013. A detailed discussion of the entire experiment with respect to ozone and bromine chemistry is given in Werner et al. (2016). SF3-2013 was a typical flight for ATTREX 2013. The GH took off from Edwards Airforce Base in California, US,





and proceeded southwards over the eastern Pacific (Figure 1 in supplement). During the initial part of the flight, from 18:00 - 19:30 UT, the GH was flying in the lower stratosphere, as illustrated by the high ozone and low $CH_4$ mixing ratios. BrO mixing ratios were $\sim 7.5 \pm 2$ ppt. $NO_2$ was between 130 - 170 ppt (Figure 13). These values are in general agreement with previous observations in the mid-latitude LS. Around 19:30 UT the GH encountered air with significantly lower ozone mixing ratios of $\sim$400 ppb and higher methane mixing ratios. BrO mixing ratios were considerably lower, $\sim 4 \pm 0.5$ ppt and $NO_2$ was between between $40 - 50$ ppt. We interpret this period, which lasted until 21:00 UT, as a flight segment in a lower part of the extra-tropical LS. Interestingly, ozone, BrO, and $NO_2$ increased again after 21:00 UT and stayed elevated until $\sim$23:00 UT. Around 23:30 UT the GH transitioned into the TTL, with a marked decrease in ozone, BrO and $NO_2$, and an increase in methane. It should be noted that the LS to TTL transition coincided with a GH descent, which makes a more detailed interpretation of this transition challenging. For the remainder of the flight, ozone mixing ratios between $17 - 17.5$ km altitude remained at $\sim 100$ppb for ozone. BrO was around $\sim 2.5 \pm 0.5$ ppt and $NO_2$ was at $\sim 15 \pm 15$ ppt.

Three descent/ascent maneuvers were performed during the second half of the flight. In all cases the GH descended to $\sim$14.5 km and climbed back to $17 - 17.5$ km altitude. The entire maneuver takes about one hour. Ozone decreased with decreasing altitude, from to $\sim 100$ ppb to $\sim 20$ ppb. Similarly, BrO decreased to $\sim 1 \pm 0.5$ ppt at 14.5 km altitude for the first two maneuvers and below the detection limit of 0.5 ppt during the third maneuver. While $NO_2$ also shows a decrease with decreasing altitude to mixing ratios close to zero, these changes are not statistically significant considering the errors of the retrievals. Figure 13 also shows a comparison with the TOMCAT/SLIMCAT model output. Because we used the ozone mixing ratios to correct the dynamical uncertainties of the model, in this case their agreement is excellent. The validity of this correction is again illustrated in the very good agreement between the methane results, with an exception early in the flight. Similarly, $NO_2$ shows excellent agreement between the observations and the model. This reflects the maturity of the ozone / $NO_x$ chemistry in TOMCAT/SLIMCAT as well as the robustness of our approach. BrO, on the other hand, shows several periods with considerable disagreements between observation and model, for example during the early part of the flight in the lower stratosphere, and during the vertical profiles. It is beyond the scope of this manuscript to discuss this disagreement, and the reader is referred to Werner et al. (2016) for more details on the interpretation of these observations.

# 6   Conclusions

We present a new small multi-spectrometer limb-scanning DOAS system developed for the use in a high altitude unmanned vehicle. The instrument was developed to fit in a small space in the GH and maintained its own environmental conditions, i.e. 0°C and vacuum, throughout the flight. The mini-DOAS was successfully deployed during the ATTREX 2013 and 2014 field deployments measuring ozone, $O_4$, BrO, and $NO_2$. Active aircraft pitch-correction reduced the pointing error in the limb scans to $\pm$0.1-0.2°, thus reducing the error in the trace gas concentration retrievals. The use of diffuser plates to measure direct solar reference spectra, which is necessary due to the low the spectral radiances in the zenith, considerably increased the limb-scanning DOAS SCD signal, and also simplified the RT calculations for the reference spectra. We believe that this scaling approach could also be useful for other aircraft instruments.





Observations were made onboard NASA's Global Hawk UAS in the tropical UT, TTL and LS between 14.5 – 17.5 km altitude. Spectral retrievals of the limb observations using established techniques yielded average SCD errors of $6.4 \times 10^{16}$ molec/cm$^2$, $2.1 \times 10^{13}$ molec/cm$^2$, and $2.6 \times 10^{14}$ molec/cm$^2$ for $O_3$, BrO, and $NO_2$, respectively. These errors are slightly larger than those from other reported aircraft DOAS systems, which are mostly operated in the cabin at pressurized and heated

conditions and optically more powerful but also heavier instrument which would not fit into the GH (Volkamer et al., 2015). The use of direct solar spectra and the resulting larger observed SCD balances the somewhat larger errors.

The mini-DOAS performed well during the ATTREX deployment, and the spectral retrieval using the diffuser plates gave high sensitivity. However, the presence of low altitude clouds and their spatial inhomogeneities poses a considerable challenge for aircraft DOAS limb observations. $O_4$ cannot be used as a proxy for the limb geometry RT in the presence of clouds. Hence

a new technique needed to be develop to overcome this challenge. It uses the scaling of limb ozone SCD with in-situ ozone observations to retrieve BrO and $NO_2$ mixing ratios. Because ozone, BrO, and $NO_2$ have similar profiles in the atmosphere, i.e. low mixing ratios in the troposphere and high mixing ratios in the stratosphere, most of the RT effects and uncertainties cancel out in the $O_3$-scaling technique. This new method allows the expansion of aircraft DOAS to cloudy conditions, which are frequently encountered and often limit the usefulness of the observations.

Considering all uncertainties of the observations and data analysis, BrO mixing ratios with errors of $\sim 0.5$ ppt can be retrieved. In the TTL the $NO_2$ mixing ratio error is below 15 ppt at all altitudes. The high sensitivities allow an analysis of the atmospheric chemistry in the UTLS at the accuracy required to provide new insights into the bromine and ozone budget. A quantitative discussion of our observation is provided in our companion paper by Werner et al. (2016).

*Acknowledgements.* This study was funded by through the NASA Upper Atmosphere Research Program (NASA ATTREX Grant numbers

NNX10AO82A for HUPCRS, and NNX10AO80A for the mini-DOAS measurements). The NOAA ozone photometer and UCATS measurements were supported by the NASA ATTREX inter-agency agreement numbers NNA11AA54I and NNA11AA55I, respectively. Additional support for the mini-DOAS measurements came through the Deutsche ForschungsGemeinschaft, DFG (through grants PF-384 5-1/2, PF384 7-1/2 PF384 9-1/2, and PF384 12-1), and the EU project SHIVA (FP7-ENV-2007-1-226224). RuShan Gao, T. D. Thornberry, and D. W. Fahey were supported by the NOAA Atmospheric Composition and Climate Program, and the NASA Radiation Sciences Program. The

TOMCAT/SLIMCAT modeling was supported by the NERC National Centre for Atmospheric Science (NCAS), UK and by the NERC TropHal project (NE/J02449X/1). We thank Dr. Eric Jensen (NASA Ames Research Center, Moffett Field, California) and his team for coordinating the NASA-ATTREX mission. We thank Joe McNorton for help with the NOAA and AGAGE CH$_4$ data. Jim Elkins of NOAA would like to acknowledge the assistance of G. S. Dutton, J. D. Nance, and B. D. Hall during the ATTREX flights, calibration, and integration.





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





**Table 1.** Instrument characteristics

| Mechanical/Electrical Properties | |
|---|---|
| Size | Cube of 45 cm side length |
| Total weight | 35 kg |
| Spectrometer assembly weight | 34 kg (includes 9 kg ice water) |
| Telescope size | 15.62 cm x 11.43 cm x 6 cm |
| Telescope weight | 0.9 kg |
| Power consumption | 28 V (DC) * 3.5 A = 100 W (including 67 W telescope heating) |
| **Telescope Optical Properties** | |
| Optical components | 12 mm lens, f $\sim$ 35 mm, 12 mm x 12 mm internal reflectance prism |
| Viewing direction | 2$°$ starboard from flight direction, actively controlled elevation selection |
| Viewing geometry | 0.25$°$ elevation, $\sim$ 1$°$ azimuth |
| Elevation aiming accuracy | better than 0.2$°$ |





**Table 2.** Spectral characteristics of the mini-DOAS spectrometers

|  | UV | VIS | Near-IR |
|---|---|---|---|
| Spectrometer | Ocean Optics QE65000 | Ocean Optics QE65000 | Ocean Optics NIRQuest 512 |
| Grating (g/mm) | 2400 | 1800 | 150 |
| Entrace slit ($\mu$m) | 200 | 100 | 100 |
| Wavelength range | 301 - 387 nm | 410 - 425 nm | 896 - 1730 nm |
| Dispersion nm/pixel | 0.085 | 0.11 | 1.6 |
| Spectral resolution | 0.6 nm | 0.8 nm | $\sim$ 15 nm |
| Filter | Hoya UG-340 | none | Schott RG 850 |
| Fiber bundle | linear, 5 times 200 $\mu$m diameter, silica | 7 times 150 $\mu$m diameter, silica | 7 times 150 $\mu$m diameter, glass |

**Table 3.** Source of absorption cross sections and reported accuracies

| Trace gas | Absorption cross section | Accuracy |
|---|---|---|
| $O_3$ | Serdyuchenko et al. (2014) | 2 - 3% |
| BrO | Fleischmann at al. (2000) | 8% |
| $NO_2$ | Bogumil et al. (2003) | 3.4% |
| $O_4$ | Thalman and Volkamer (2013) | 3% |
| $H_2O$ | HITRAN 2012 | 8% |



**Table 4.** Details of the spectral analysis of various trace gases

| Species | Wavelength range (nm) / pixel | Trace gases fitted | Average DSCD error (molec/cm$^2$) / molec$^2$/cm$^5$ for O$_4$) |
|---|---|---|---|
| **UV** | | | |
| O$_3$ | 346 − 354.7 / 552 − 665 | O$_3$, O$_4$ (293 K), NO$_2$, BrO | $6.4 \times 10^{16}$ |
| BrO | 346 − 360.3 / 552 − 740 | BrO, O$_3$ (203 K), O$_3$ (213 K, orthogonalized to low T O$_3$),O$_4$, NO$_2$ (203K) | $2.1 \times 10^{13}$ |
| O$_4$ | 338.8 − 366.3 / 460 − 820 (excluded 347.4 − 352 / 570 - 820) | O$_4$ (293 K), BrO, O$_3$ (203 K), O$_3$ (213 K, orthogonalized to low T O$_3$), NO$_2$ (203K) | $7.1 \times 10^{40}$ |
| **VIS** | | | |
| O$_3$ | 437.1 − 485.5 / 250 − 720 | O$_3$ (203 K), O$_4$ (293 K), NO$_2$ (203 K), NO$_2$ (223 K, orthogonalized), H$_2$O (203K) | $6.2 \times 10^{17}$ |
| NO$_2$ | 424.2 − 460.4 / 130 − 470 | O$_3$ (203 K), O$_3$ (223 K, orthogonalized to low T), O$_4$ (293 K), NO$_2$ (203 K), H$_2$O (203 K) | $2.6 \times 10^{14}$ |
| O$_4$ | 459.8 − 488.4 / 464 − 750 | O$_3$ (203 K), O$_3$ (223 K, orthogonalized to low T), O$_4$ (293 K), NO$_2$ (203 K), H$_2$O (203 K) | $2.8 \times 10^{41}$ |





**Table 5.** Sensitivity runs for inferred $NO_2$, and BrO, according to the $O_3$-scaling technique.

| Run # | Parameter | Modification | Δ $NO_2$ | | | |
| --- | --- | --- | --- | --- | --- | --- |
| | | | absolute | | relative[%] | |
| | | | range | typical | range | typical |
| 1 / 2 | In-situ$O_3$ | × 0.97/1.03 | ± 10 ppt | 4/2 ppt | ±6 | ± 3 |
| 3 / 4 | $NO_2$ profile > 17.5 km | × 0.85/1.15 | ± 9 ppt | ±3 ppt | ± 3 | ± 5 |
| 5 / 6 | | × 0.7/1.3 | ± 25 ppt | ±13 ppt | ±30 | ± 15 |
| 7 / 8 | $O_3$ profile >17.5 km | × 0.9/1.1 | ± 9 ppt | ±3 ppt | ±18 | ± 13 |
| 9 / 10 | | × 0.97/1.03 | ± 3 ppt | ±1.5 ppt | ±5 | ±3 |
| 11, 12, 13 | Tropospheric $NO_2$ | + 10/+ 15/+ 20 ppt | ±15/±20/±50 ppt | ±10/±15/±25 ppt | ±1000/±1500/±2000 | ±500/±750/±1000 |
| 14 / 15 | Aerosol extinction | × 0.5/2 | ±4/±2 ppt | ±1.5/±1.5 ppt | ±4/±4 | ±3/±3 |
| 16, 17, 18 | Marine strato-cumulus | OD=5/10/20 from 1-2 km | ±3.5/±3.5/±3.5 ppt | ±1.5/±1.5/±1.5 ppt | ±4/±5/±5 | ±5/±6/±7 |
| 19, 20, 21 | Cirrus cloud | OD = 1 from 13-14/ 14-15/15-16 km | ±5/±5/±35 ppt | ±3/±3/±3 ppt | ±20/±30/±50 | ±5/±15/±20 |
| 22, 23, 24 | Visible ground albedo | A = 0.1/0.3/0.4 | ±3/±4/±6 ppt | ±2/±2/±2 ppt | ±2/±3/±4 | ±2/±2/± |
| 25 / 26 | Pointing error | ± 0.2° | ±10/±3 ppt | ±4 ppt | ±30 | ±30/±3 |

| Run # | Parameter | Modification | Δ BrO | | | |
| --- | --- | --- | --- | --- | --- | --- |
| | | | absolute | | relative[%] | |
| | | | range | typical | range | typical |
| 1 / 2 | In-situ $O_3$ | × 0.97/1.03 | ±0.2 ppt | ±0.15 ppt | ±4 | ±3 |
| 3 / 4 | BrO profile > 17.5 km | × 0.85/1.15 | ±0.6 ppt | ±0.2 ppt | ± 30 | ± 10 |
| 5 / 6 | | × 0.7/1.3 | ±0.6 ppt | ±0.2 ppt | ± 30 | ± 10 |
| 7 / 8 | $O_3$ profile >17.5 km | × 0.9/1.1 | ±0.4 ppt | ±0.2 ppt | ± 8 | ± 5 |
| 9 / 10 | | × 0.97/1.03 | ±0.2 ppt | ±0.1 ppt | ± 4 | ± 3 |
| 11, 12, 13 | Tropospheric BrO | + 0 ppt, + 1/+ 1.5 ppt | ±0.5/±0.5/±1 ppt | ±0.3/±0.3/±0.6 ppt | ±70/±50/±100 | ±15,±8,±20 |
| 14 / 15 | Aerosol extinction | × 0.5/2 | ±0.04/0.06 ppt | ±0.03/±0.015 ppt | ±1/±1 | ±0.5/±0.5 |
| 16, 17, 18 | Marine strato-cumulus | OD=5/10/20 from 1-2 km | ±0.1/±0.2/±0.3 ppt | ±0.07/±0.1/±0.25 ppt | ±4/±6/±8 | ±3/±4/±5 |
| 19, 20, 21 | Cirrus cloud | OD = 1 from 13-14/ 14-15/15-16 km | ±0.1/±0.3/±0.5 ppt | ±0.07/±0.15/±0.20 ppt | ±10/±35/±25 | ±4/±6/±8 |
| 22, 23, 24 | UV ground albedo | A = 0/0.1/0.2 | ±0.05/±0.05/±0.1 ppt | ±0.025/±0.025/±0.05 ppt | ±1/±1/±3 | ±0.5/±0.5/±2 |
| 25 / 26 | Pointing error | ± 0.2° | ±0.3 ppt | ±0.1/±0.3 ppt | ±17 | ±3/±15 |





**Figure 1.** Sketch of the mini-DOAS instrument, as developed for NASA's Global Hawk.



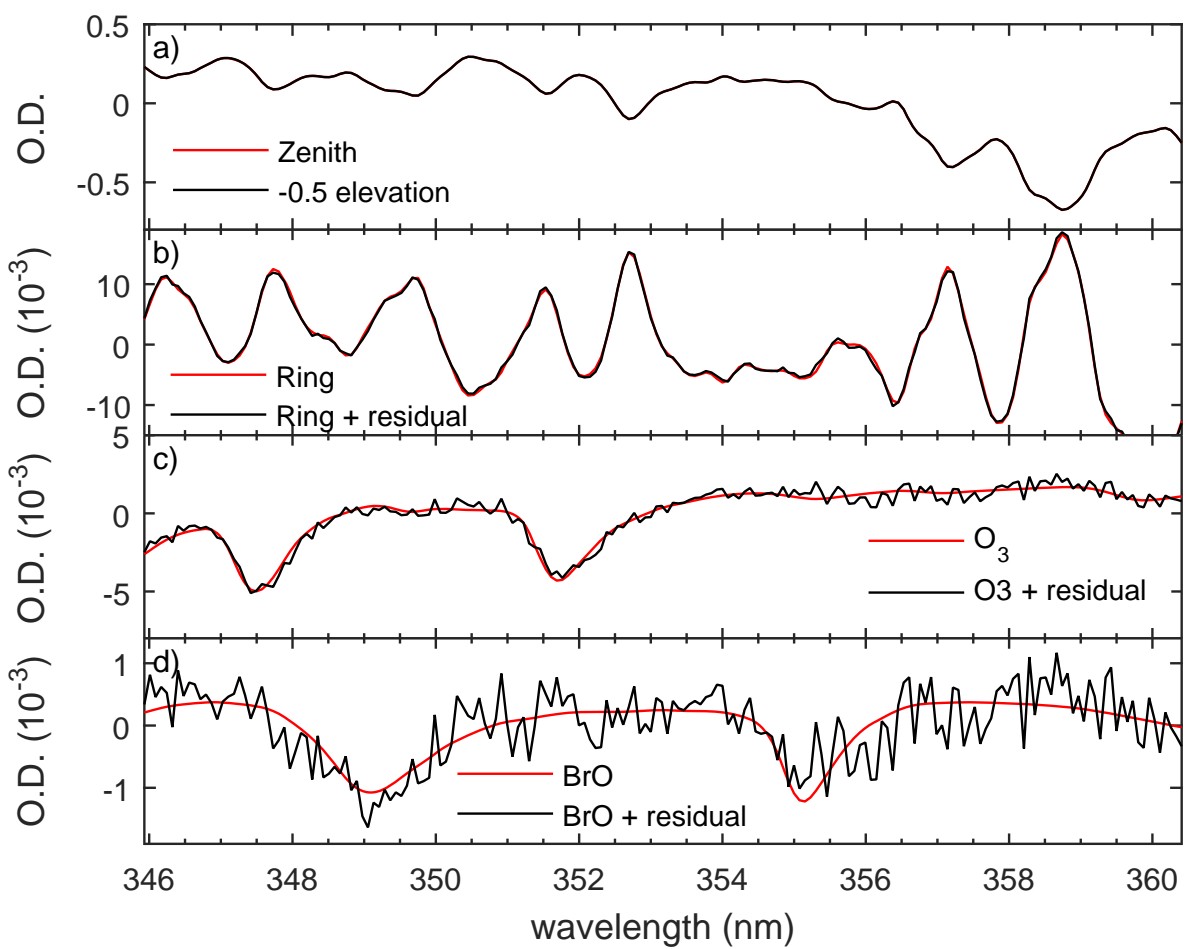

**Figure 2.** Example of the spectral analysis of BrO during SF3-2013 (Feb. 14, 2013) at 20:28 UT at an altitude of 17.0 km at 29.8°N, 128.9°W. The original spectrum was recorded with an elevation viewing angle of -0.5°. Optical densities, O.D., are in arbitrary units. The top panel compares the -0.5°spectrum to the solar reference, while the second panel shows that the Ring spectrum is the second most prominent spectral feature in this spectrum. Ozone was retrieved using two reference spectra for different temperatures. Panel c shows the sum of these two absorption spectra compared to this sum added to the fit residual. Panel d shows the comparison of the the absorption of BrO for a DSCD of $(1.94 \pm 0.19) \times 10^{14}$ molec/cm$^2$. The two BrO absorptions bands are clearly identified in this analysis.



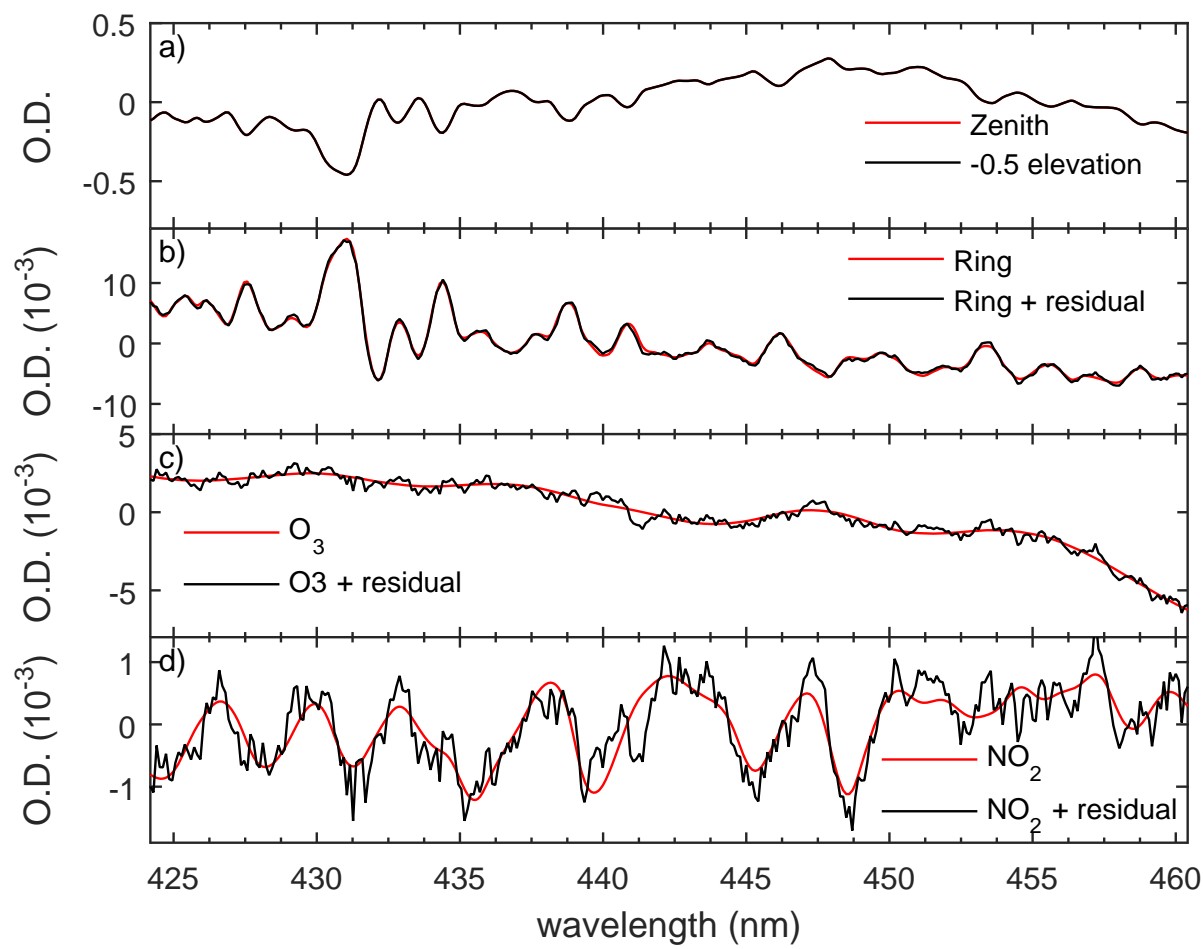

**Figure 3.** Example of the spectral analysis of $NO_2$ during SF3-2013 (Feb. 14, 2013) at 20:57 UT at an altitude of 17.1 km at 28.2°N, 131.2°W. The original spectrum was recorded with an elevation viewing angle of -0.5°. Optical densities, O.D., are in arbitrary units. The top panel compares the -0.5° spectrum to the solar reference, while the second panel shows the fitted Ring spectrum. Panel c shows the sum of two ozone absorption spectra at different temperatures compared to this sum added to the fit residual. Panel d shows the comparison of the the absorption of $NO_2$ for a DSCD of $(5.38 \pm 0.22) \times 10^{15}$ molec/cm$^2$. Multiple $NO_2$ absorptions bands are clearly identified in this retrieval.



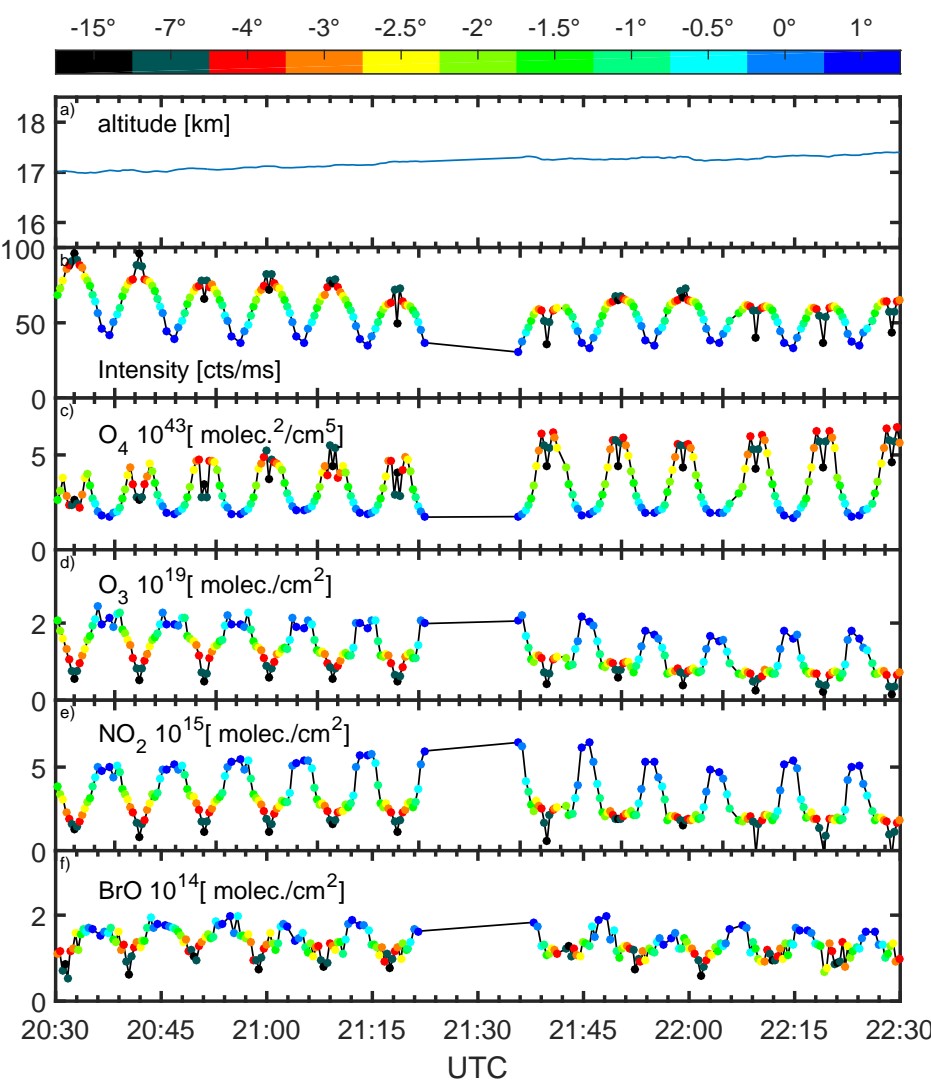

**Figure 4.** Overview of a 2 hour segment of SF3-2013 (Feb. 14, 2013) with varying cloud cover: 20:30 - 20:45 UT clouds between 11 -12 km; 20:45-21:10 UT clouds between 7 - 9 km; 21:45 - 22:15 UT few clouds between 1 - 2 km altitude. The different colors denote different elevation angles (see color scale at the top). Intensity, $O_4$, $O_3$, and $NO_2$ DSCDs were derived in the visible wavelength range, while the $BrO$ DSCD was retrieved in the UV.



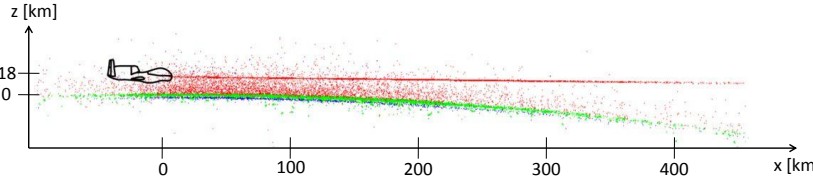

**Figure 5.** 3-D simulation of the radiative transfer of the limb measurement at 18 km altitude above a marine strato-cumulus cloud deck located at 2 km altitude with optical thickness of 30. Red, green and blue points mark each Rayleigh, Mie, and ground reflection scattering event, respectively.

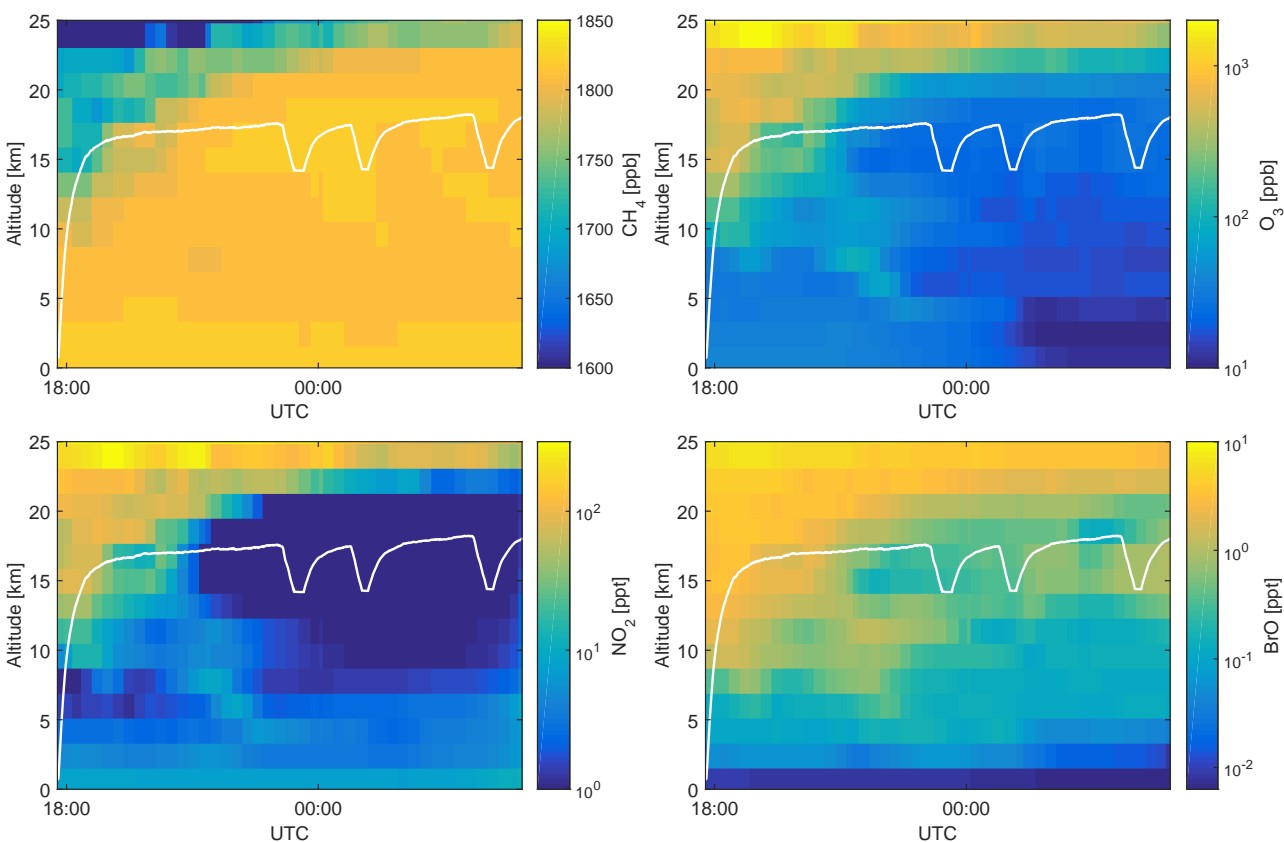

**Figure 6.** TOMCAT/SLIMCAT simulation of mixing ratio curtains (on a log scale) of $CH_4$, (upper left panel), $O_3$ (upper right panel), $NO_2$ (lower left panel), and BrO (lower left panel) together with the flight trajectory for the sunlit part of SF3-2013 (Feb. 14, 2013). For better visibility, the simulated mixing ratios are shown for the altitude range 0 - 25 km, although the TOMCAT/SLIMCAT simulations cover 36 unevenly spaced levels between 0 - 63 km altitude.





**Figure 7.** Altitude dependent contribution to the simulated optical densities of the $O_2$ - $O_2$ collisional complex ($O_4$) at 360 nm for limb measurements at 18 km and different observation angles, as indicated in the legend of panel (a). In the simulations, a deck of marine strato-cumulus clouds (mSc) located between 1 - 2 km of different cloud optical depth $\tau_{mSc}$ is assumed, since according to the cloud physics Lidar measurements (CPL). mSc clouds were frequently occurring during the NASA ATTREX flights over the Eastern Pacific. Panel (a) is for clear skies, panel (b) for $\tau_{MSc} = 1$, panel (c) for $\tau_{mSc} = 5$, panel (d) for $\tau_{mSc} = 10$; panel (e) for $\tau_{mSc} = 20$, and panel (f) for $\tau_{mSc} = 50$. The integral under the curves corresponds to the optical density an observer would measure for the given conditions.




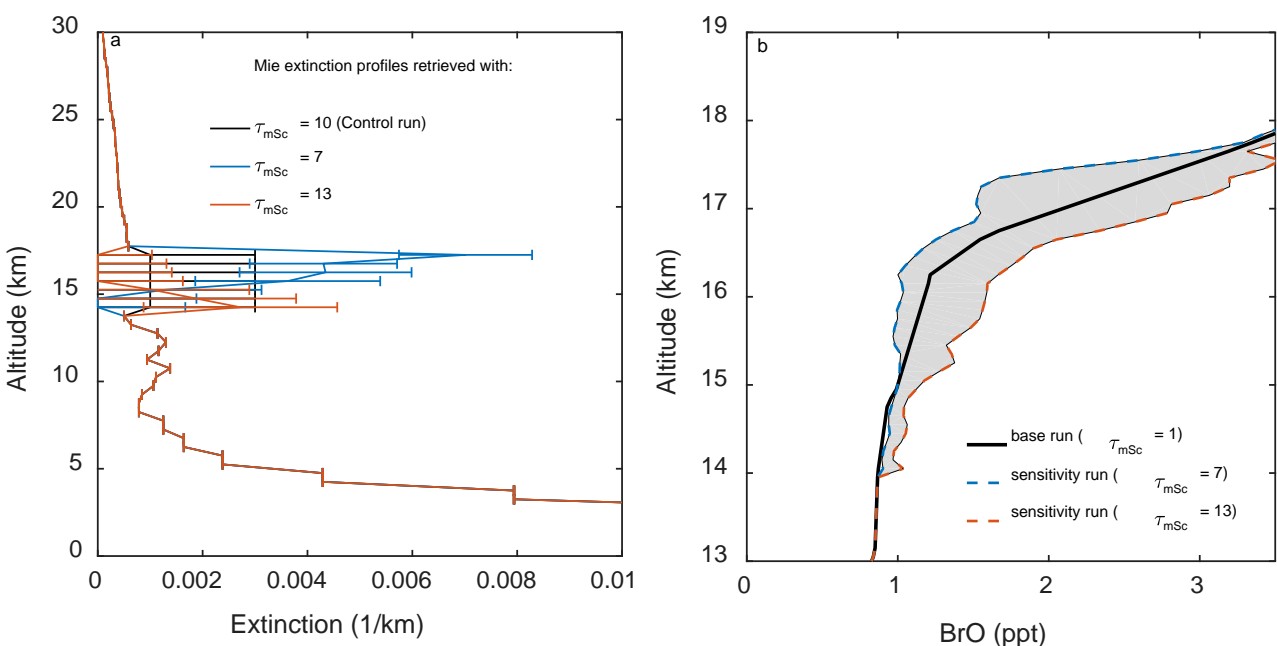

**Figure 8.** Theoretical study of the sensitivity of the aerosol and BrO optimal estimation retrieval on low clouds. The left panel shows the theoretical retrieval of an aerosol extinction profile from a theoretically calculated $O_4$ vertical profile for a marine cloud with $\tau_{mSc} = 10$ (control run, black curve), when the optical depth of the cloud in the retrieval is assumed to be $\tau_{mSc} = 7$ (blue curve) and $\tau_{mSc} = 13$ (red curve). The right panel shows how the variation on retrieved optical depth is propagated onto the BrO profile retrieval.





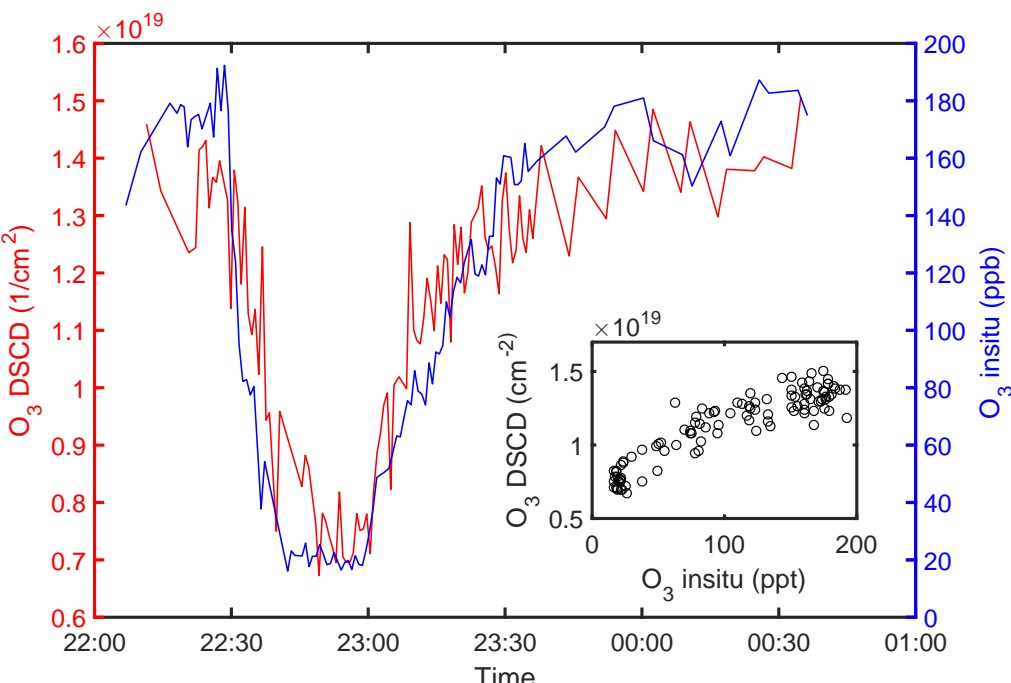

**Figure 9.** Comparison of limb $O_3$ DSCD and in-situ mixing ratios during the second vertical profile maneuver during SF3-2013. The insert shows the clear correlation between the mini-DOAS observations and the flight-level ozone mixing ratios, which indicates the high sensitivity of the mini-DOAS at flight level.





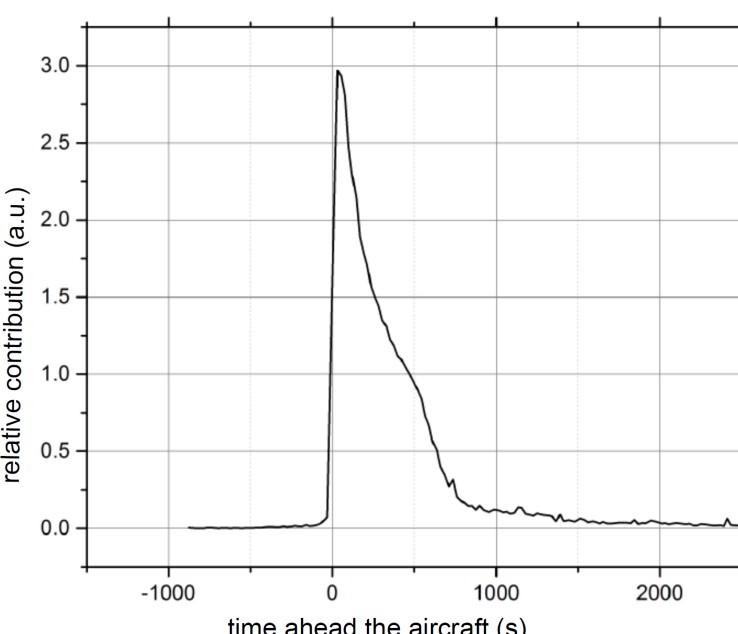

**Figure 10.** Relative contribution to the analyzed light as function of the time ahead of the actual flight position as predicted by 3D simulation of the radiative transfer (Raecke, 2013).





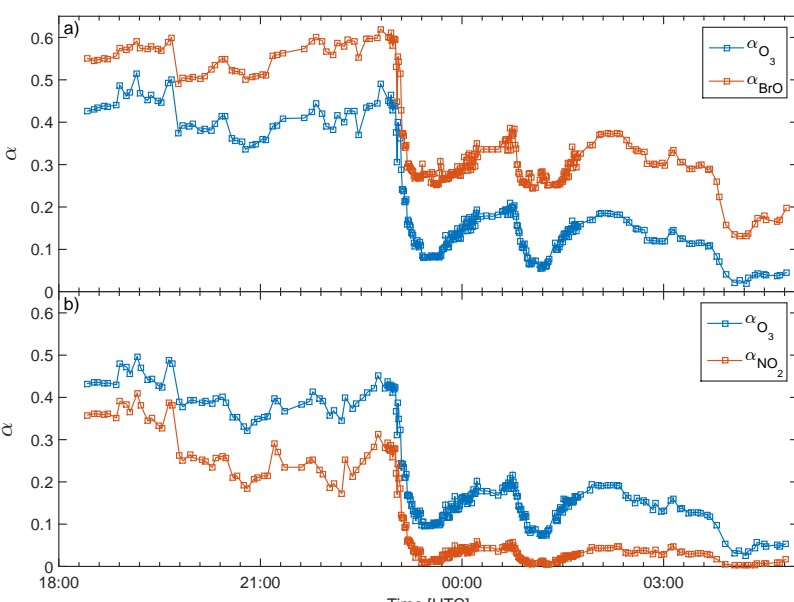

**Figure 11.** $\alpha$-factors (equations 11, and 13) for measurements of $O_3$, and BrO in the UV spectral range (panel a), and $O_3$ and $NO_2$ in the visible spectral range (panel b) (for the wavelength ranges see Table 2) for the sunlit part of SF3-2013 (Feb. 14, 2013).



**Figure 12.** Panel (a) shows the time altitude trajectory of the sunlit part of SF3-2013 (Feb. 14/15, 2013). Inter-comparison of measured and TOMCAT/SLIMCAT-simulated $CH_4$ (HUPCRS) (panel b), $O_3$ (NOAA) (panel c), $NO_2$ (mini-DOAS) (panel d), and BrO (mini-DOAS) (panel e). The errors of the mini-DOAS $NO_2$, and BrO, shown as grey areas in panels (d) and (e), include all dominating errors, i.e., the spectral retrieval error, the overhead and the error due to a tropospheric contribution to the slant absorption. The dashed lines in panels (d) and (e) represent the detection limits of the mini-DOAS observations.







**Figure 13.** Same as 12, but forcing NOAA-measured $O_3$, HUPCRS-measured $CH_4$ to agree with TOMCAT/SLIMCAT-modelled prediction in a least square sense. The forcing is performed by interpolating the predicted profile of the gases, and subsequent vertically shifting the whole packages of predicted gases, until measured and modeled $O_3$, and $CH_4$ agree best. Typical vertical shift range from 0 to 1000 m, the latter being about the vertical spacing of the TOMCAT/SLIMCAT levels. The errors of the mini-DOAS $NO_2$, and BrO are shown as grey areas in panels (d) and (e). The dashed lines in panels (d) and (e) represent the detection limits of the mini-DOAS observations.





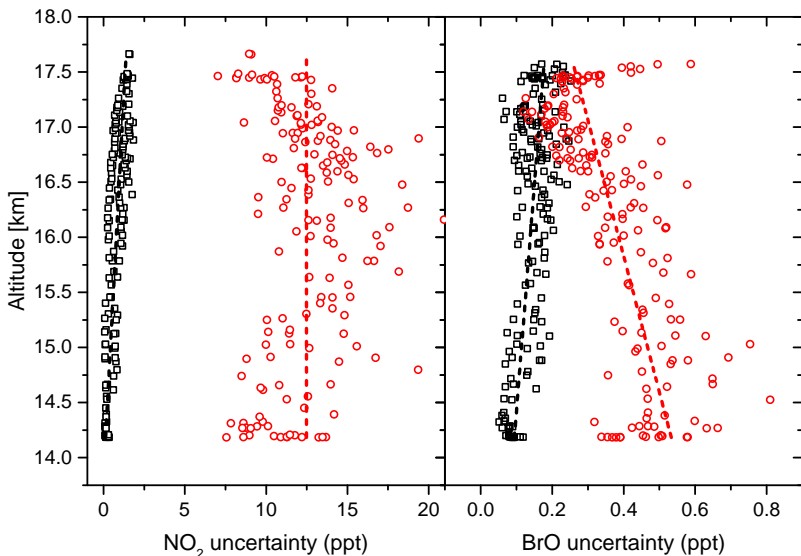

**Figure 14.** Uncertainty in the inferred $NO_2$ (left panel) and BrO (right panel) as a function of altitude due to uncertainties in the overhead (in black), and below the aircraft located column amounts (in red) of the respective gas. For the overhead column amounts, uncertainties of $\pm$ 15% were assumed for both $NO_2$, and BrO. The uncertainty due to below the aircraft located $NO_2$, and BrO is estimated by assuming uniform tropospheric mixing ratios of 15 ppt, and 0.5 ppt, respectively (for details see section 4.4). The dashed lines indicate the linear regressions, which was used in the calculation of error propagation.





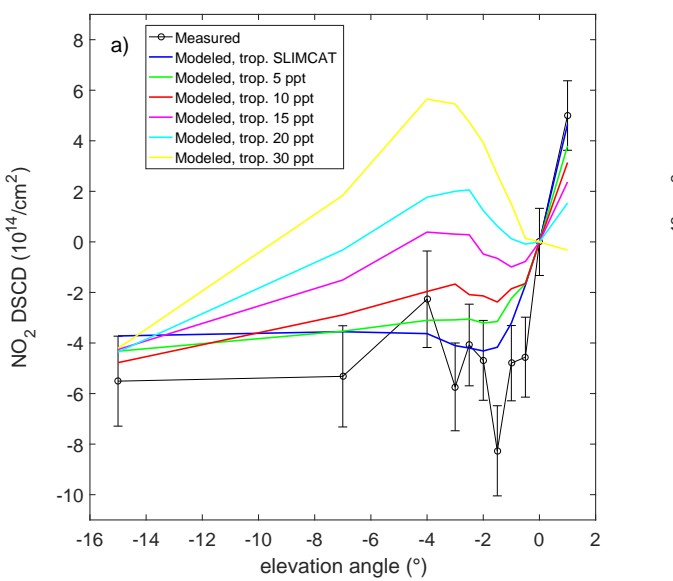
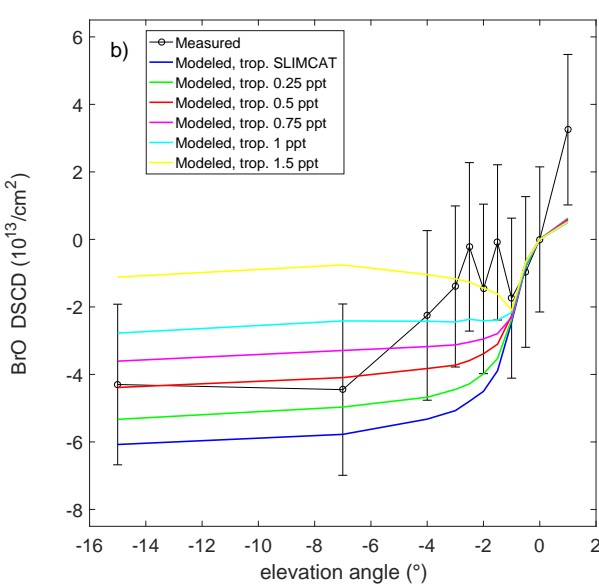

**Figure 15.** Sensitivity study to determine tropospheric (a) $NO_2$ and (b) BrO mixing ratios. The black curve shows the $NO_2$ and BrO SCD from an elevation scan performed during the first descent of SF3-2013. These values are compared to various forward calculations based on TOMCAT/SLIMCAT output for this specific flight segment and various assumed tropospheric $NO_2$ and BrO mixing ratios. Within the uncertainty of the observations, the tropospheric $NO_2$ can be determined to be below $\sim 10$ppt, while BrO is around $0.5 \pm 0.5$ ppt and possibly somewhat larger ($\leq 1.5$ ppt) just below the flight altitude.