# Peer review of "A New Differential Optical Absorption Spectroscopy Instrument to Study Atmospheric Chemistry from a High-Altitude Unmanned Aircraft"

_Atmospheric Measurement Techniques, 2016_

## Referee Comment (RC1) · Anonymous Referee #2 · 23 Dec 2016

**Referee Comment on amt-2016-251 by J. Stutz et al., 2016**

The manuscript describes a new DOAS (Differential Optical Absorption Spectroscopy) instrument which is deployed on NASA's Global Hawk UAS (unmanned arial system) and measures scattered sunlight in the UV and vis wavelength regions in limb scanning geometry. Analyses of atmospheric trace gases are performed at altitudes between the upper troposphere and the lower stratosphere.
The setup of the instrument, its viewing geometry, as well as the purpose and benefits

of the specific methods are described and presented. In addition, first measurement results of several trace gases are presented and discussed. Derivation of mixing ratios from DOAS observations is a challenge especially at high altitudes. The authors use and describe a new method to derive BrO and $NO_2$ mixing ratios at high altitudes from the observed slant columns by using in-situ ozone observations for scaling purposes. Many aspects including the error influences of several model assumptions are analysed in detail. Deployed on the GH on a frequent basis, the instrument may yield valuable insights into and better understanding of the lower stratospheric abundances of $O_3$, BrO and $NO_2$. Reference to a companion paper is given in a suitable way. The content and significance of the manuscript are well within the scope of AMT.

Most aspects are well explained and the quality of the measurements and analysis is good. However, the text would benefit in some parts if the authors put some more carefulness in the writing, i.e. the phrasing and correct grammar. There are quite a number of mistakes/oversights such as missing words, as well as a few passages which are not so well described. This makes the reading somewhat cumbersome in the respective places, while the content itself is mostly fine. Please check the text once more in addition to the comments given here. With some additional effort on the text and some clarifications, the manuscript is certainly suitable for publication in AMT.

**I. Main Comments**

**Comment 1. Section 4.2, l. 1059**
Please describe and/or cite what is used as "typical BrO concentration profile". It is also unclear how the choice of this influences the uncertainty determination.

**Comment 2. Section 4.3, l. 1202-1208**

Please give some more clear explanation on how (strong) the model assumptions and especially the finally applied values of $SCD_{ref}$ and the vertical trace gas profiles from the TOMCAT/SLIMCAT model, which constrain the RT simulations, influence the mixing ratio results received from the $O_3$-scaling method. This influence is also mentioned further below (Section 4.3, l. 1242). It becomes clear that certain effects and assumptions cancel in the applied method but others don't. It is also clear from the detailed descriptions and equations that this input information is needed. However, it is necessary to understand in how far the results and resulting interpretations from the announced new scaling method are independent from the initial assumptions, especially $SCD_{ref}$ and the vertical profiles.

The Supplement gives a wide range of valuable information on influences from the model assumptions on the final results, as summarized in Table 5. However, $SCD_{ref}$ is not considered and the vertical profiles are only scaled (either for the tropospheric or the stratospheric part) but not changed in their actual shape. If this (i.e. absolute changes as opposed to the shape in the two separate altitude ranges) is representative for the entire profile influence on the final results this should be stated more clearly. One could imagine that especially relative changes in the profile shapes between ozone and $NO_2$ or BrO are relevant for the final mixing ratio results. Please add some explanations as clarification.

**Comment 3. Section 4.3, ll. 1269-1303, Figures 12 and 13**

In part this comment is connected to Comment 2. It is not clear how the shifting procedure is performed. At first reading the procedure seems easy. However, comparing with Figures 12 and 13 some questions arise. First of all, the caption of Figure 13 is really misleading. It states that the measurement results are forced to match the model results, which according to text and figures should be vice versa. The measurement curves are the same as in Fig. 12, and the model results change from Fig. 12 to Fig. 13. As stated in the text, the model is forced to match the measurements, correct?

The change in ozone between pre and post shifting in some parts of the flight, however, is quite large up to a factor of two it seems (e.g. between 18:00 and 20:00 UTC), and maybe around 10-20% in other parts of the flight. The text, however, states that the vertical shift needed for matching the curves is usually smaller than 1km. Is the ozone changing that strongly between one model layer and the next? Maybe there is some misunderstanding. Maybe it would be also useful to give some more information on the used vertical profiles in general, e.g. in the Supplement, see also Comment 1. In addition: Why is a vertical shift sufficient? Is this simply because of monotonically increasing concentrations? Please briefly clarify.

**Comment 4. Section 6**
As outlook and for a better understanding of how the measurements performed by the new DOAS instrument on the UAS will yield further information on the UTLS, it would be very helpful if a statement - e.g. at the end of the conclusions - would be given on the future plans, especially on how often and for how long etc. the instrument can - and will be - used for the type of measurements presented here, and for which further research the results will be helpful.

**II. Minor Comments**

1. Lines 19-20: It is somewhat strange to concentrate on relative errors here. In order to judge the performance of the instrument, absolute errors are in addition helpful.
2. Line 57: Please state and cite actual values (current best knowledge) of total amounts of $Br_y^{org}$ and $Br_y$ so the given numbers can be directly set into relation.
3. Line 87: Is 3ppt the current uncertainty or the goal for the improved uncertainty of $Br_y$? Please state the respective other quantity. (The newly achieved uncertainty would be interesting, however, according to the grammar of the sentence the given

value is the current uncertainty. Please clarify.)

4. Line 97: Earliest measurements of? Does this refer to BrO?

5. Lines 305-307: Please give a short reason (too dark due to lack of scattered sunlight?).

6. Lines 388-390: This does not really explain how the opening angles are actually determined. The scanning capability obviously helps in the calibration. Please shortly state how the FOV was measured.

7. Lines 425-428: Although this range of the telescope stability is within the uncertainty of the elevation calibration, it obviously adds to the overall uncertainty of the determined elevation. Is this included in the uncertainty considerations? It doesn't sound as if it is. Or maybe the phrasing is misleading?

8. Lines 457-461: What about the roll angle? Probably it is just missing in the list.

9. Lines 549-551: A common shift would not be an error of the trace gas reference spectra which are all from individual data, but a (very small in the presented case) uncertainty from the instrument calibration which seems to be fine.

10. Line 556: Please refer to Table 3.

11. Lines 567-570: Reading this remark, the reader asks for an explanation, which is only given much later. Either shortly state or refer to Section 4.

12. Lines 575-578: The excluded range lies within the fitting windows for $O_3$ and BrO also, but is not excluded there, right? A short comment of reasoning would be helpful. (Why does the feature lead to a problem only in the $O_4$ fit?)

13. Line 608: Please give the ozone window also in the text as for the other trace gases, not only in the table.

14. Line 644: New paragraph?

15. Lines 698-705: Please check this passage, there is some inconsistency. If the reduction in $O_3$ and $NO_2$ is "not as pronounced" it would mean that at EA 1° the DSCD is similar as in the lower EA, so the relative error for $O_3$ and $NO_2$ would be even more problematic if using EA 1° as reference instead of direct sun. Please clarify.

16. Lines 749 and 763: Is it true that one instrument/method gives relative errors and the other one absolute errors only? Otherwise please use comparable data/information.

17. Line 792: Is an albedo of 0.2 realistic for the VIS? This seems rather large.

18. Lines 845-849: Please briefly state which observation leads to the interpretation (transition from lower stratosphere to UTLS region).

19. Lines 1006-1007: This information should be given earlier in the section, i.e. at first mentioning.

20. Lines 1389-1391: As the sentence can be read in a somewhat misleading way, a recommendation would be a rephrasing of the following kind: "As further analysis goes beyond the scope of this manuscript, we only briefly discuss the results (...)."

21. Lines 1485-1486: (...) i.e. low mixing ratios in the troposphere (...). This is, however, not generally true for $NO_2$ at all. This statement requires some clarification.

22. Lines 1216-1219 and Figure 10: Does the calculation depend on certain experimental flight conditions? Just trying to understand why there are kinks in the curve, e.g. at 700s. This is not clear. Please give a short explanation.

**III. Technical and text/grammar corrections**

1. Line 5-9: (...) developed for (...) GH (...) _and tested/applied_ during (...) ATTREX.

2. Line 26: (RT) not needed in abstract, abbreviation should be given in main text (Line 199?).

3. Line 151: (...) interpretation _of_ the data.

4. Line 159: Sounds as if the name of the aircraft is missing.

5. Line 181: New paragraph?

6. Line 193: Please add abbreviation DSCD at the end of the sentence as definition.

7. (...) the side facing _of_ the diffusers (...)

8. Line 541: The term "dispersion" seems to be the wrong choice. Do you rather mean

"convolution"?

9. Line 673: (...) and in _the/a_ companion paper (...)

10. Lines 770-773: Sentence misses a verb. E.g. (...) we also employ a 3D atmospheric chemistry model to _determine_ vertical trace gas concentration profiles.

11. Lines 813-816: Although (probably) not wrong, the sentence structure makes it really hard to understand the meaning, especially the subject of the sentence "TOMCAT/SLIMCAT model meteorology".

12. Line 832: Remove the space at the end of the sentence.

13. Lines 860-861: (...) at the time and location _of_ the measurement (...).

14. Lines 987-989: Missing word "and": (...) _and_ a non-linear retrieval (...) is employed.

15. Line 996: The solution of this iteration is _a_ (instead of "an") vertical (...) profile (...).

16. Line 1052: (...) the general error of _a_ (instead of "an") non-linear extinction (...)

17. Lines 1077-1082: Devide into two sentences please.

18. Line 1081: "than" instead of "that"

19. Line 1124: typo in "predomina_n_tely"

20. Line 1125: (...) the ratio of the two parameters _is_ (instead of "are") a measure (...).

21. Line 1140-1142: Please correct this sentence.

22. Line 1144: Due _to_ (...)

23. Line 1171: "B" should be in italic format.

24. Lines 1269-1270: (...) requires addition_al_ scrutiny (...)

25. Line 1457: Please delete the extra "the".

26. Line 1460-1461: Apparently this sentence belongs to a different paragraph. It does not fit to the content here at all, but probably should rather be shifted to line 1490.

27. Lines 1470-1471: The verb "are" needs to be repeated here at least once: (...) and _are_ optically more powerful (...).

28. Line 1481: (...) needed to be develop_ed_ (...).

[Figure]

29. Table 2: The wavelength range for the VIS instrument really seems to be wrong. The given range from 410-425 nm would not cover the retrieval windows given in Table 4.

30. Table 4: Please correct the title of the last column. e.g. Average DSCD error (molec/cm$^2$, for $O_4$: molec$^2$/cm$^5$) or similar.

31. Figure 2 and 3: $O_3$ (please use subscript).

32. Caption of Figure 2: delete second "the" in second of last line.

Missing plural "s", e.g., at the following locations:

Line 153: Few research aircraft_s_

Line 158: (...) a number of aircraft_s_

Line 214: Theoretical consideration_s_

Line 1046: (...) these result_s_

Line 1290: (...) the modeled $O_3$ profile_s_ are (...)

Line 1326: (...) spectral retrieval error_s_ (...) which are (...)

Line 1471: (...) instrument_s_

Corrections w.r.t. references and citations:

1. Line 40 (and many other locations): The format of references / citations has some issues. Please remove the extra parentheses around the year of publication.

2. Line 163: Parentheses around the two references are missing.

3. Please correct typo in name Baidar in Baidar et al., 2013.

Corrections concerning the Supplement:

Text correction in the figure captions: (...) parameters given _in_ Table 5.

---

## Referee Comment (RC2) · Anonymous Referee #1 · 6 Jan 2017

This manuscript describes a new and novel DOAS instrument for observations of trace gases from an unmanned aircraft. The article is generally readable, although the language is a bit conversational in places, and fully explores issues with these types of measurements. The methods compare constraint of the radiative transfer model via O4 observations to a novel O3 scaling method and demonstrate that use of O4 is challenging and the O3 method resolves many of these issues. The work clearly demonstrates an advance in airborne DOAS observations, including the first observations from a high-altitude unmanned platform and is appropriate for publication in AMT.

[Figure]

Specific Comments:

page 1, line 12: The wording of "due to frequent presence of low altitude clouds" is not very clear here as the sentence is talking about O4, not clouds. The wording could be made more clear.

page 1, line 15: I believe the "precision" of the measurement is being discussed, not the "accuracy"

page 2, line 10: Bry is certainly common, but should be defined here. The text defines it as a sum of organic and inorganic Bry, but fails to define Bry in general.

page 2, line 22: Photolysis rate of BrO (JBrO) also affects partitioning.

page 3, line 14: I don't think "nadir" is all caps. Also, these column-sensitive instruments "...provide no _altitude specific_ information on the UTLS."; they quantify the column including the UTLS contribution to the column.

page 3, line 22: Fix wording: "...interpretation of the data..."

page 4, line 16: Multiple parameters can be used for aerosol extinction constraint (e.g. use of radiances and O4 observations).

page 4, line 18: O4 does vary in time (as pressure and to a lesser extent temperature vary).

page 5, line 14: The citation to ground-based MAX-DOAS systems should include an "e.g." because these are a small subset of those references. Also, Hoenninger et al. (doi:10.5194/acp-4-231-2004, 2004) should be cited here as a seminal paper in ground-based MAX-DOAS.

page 7, line 3: How many fibers in bundle? Optical core diameters? This is in the table, so a reference to the table (2) can work.

page 8, line 8: I believe it should say "low solar zenith angle" rather than "...elevation"

page 9, line 8: This is a slant path for the integration; make more clear.

page 13, line 18: The albedo of ocean (0.2 in visible) seems high. Why is this chosen? Reference?

page 16, line 3: Replace "it" with "O4 reference", and indicate "for general (e.g. not fully clear sky) conditions."

page 18, line 13: There are three citations to theses in this line, and in general these theses may not always be accessible, nor are they fully peer reviewed. Other instances of theses also exist in the citations. It should be made clear that this method is described in those theses, but was not published in peer-reviewed form in those theses. See AMT reference guidelines.

page 20, line 33: Please also show the ratio of alpha factors to demonstrate the degree of "cancellation" of the variability of the individual factors.

page 22, line 7: I believe the column of overhead ozone is varied, not the shape of the profile. Please clarify.

---

## Author Comment (AC1) · 10 Feb 2017

**Response to Reviewers Comments on amt-2016-251: A New Differential Optical Absorption Spectroscopy Instrument to Study Atmospheric Chemistry from a High-Altitude Unmanned Aircraft, by Stutz et al.**

We would like to thank both reviewers for their helpful comments. Below are our responses (in italic) to the comments.

**Response to Reviewer 1:**

page 1, line 12: The wording of "due to frequent presence of low altitude clouds" is not very clear here as the sentence is talking about O4, not clouds. The wording could be made more clear.

*Wording was changed to: "This is due to the frequent presence of low altitude clouds, which shift the sensitivity of the $O_4$ SCD into the lower atmosphere and make it highly dependent on cloud coverage."*

page 1, line 15:  I believe the "precision" of the measurement is being discussed, not   the "accuracy"

*No, this is indeed the accuracy, as explained in detail in section 4.4. However, we changed the value to ~0.5 ppt, to make it consistent with Section 4.4 (this was changed in the AMTD editor review in the text but had been overlooked in the abstract).*

page 2, line 10: Bry is certainly common, but should be defined here. The text defines it as a sum of organic and inorganic Bry, but fails to define Bry in general.

*To be more clear, this was changed to : "…total concentration of organic and inorganic bromine…"*

page 2, line 22: Photolysis rate of BrO (JBrO) also affects partitioning.

*That is correct. The following words were added to this sentence: "…, as well as the actinic flux."*

page 3, line 14: I don't think "nadir" is all caps. Also, these column-sensitive instruments "...provide no _altitude specific_ information on the UTLS."; they quantify the column including the UTLS contribution to the column.

*Nadir is now in lower case and text has been changed to "provide no altitude specific information on the UTLS"*

page 3, line 22: Fix wording: "...interpretation of the data..."

*corrected*

page 4, line 16: Multiple parameters can be used for aerosol extinction constraint (e.g. use of radiances and O4 observations).

*Changed "parameter" to "parameters"*

page 4, line 18: O4 does vary in time (as pressure and to a lesser extent temperature vary).

*That is correct. Text was changed to : "…well-defined atmospheric profile."*

page 5, line 14: The citation to ground-based MAX-DOAS systems should include an "e.g." because these are a small subset of those references. Also, Hoenninger et al. (doi:10.5194/acp-4-231-2004, 2004) should be cited here as a seminal paper in ground-based MAX-DOAS.

*Done. Changed to : "...(e.g. Hönninger et al., 2004; Pikelnaya et al., 2007; Platt and Stutz, 2008)."*

page 7, line 3: How many fibers in bundle? Optical core diameters? This is in the table, so a reference to the table (2) can work.

*To make sure the reader can find the information we changed the text to: "...(see Table 2 for details)"*

page 8, line 8: I believe it should say "low solar zenith angle" rather than "...elevation"

*Thanks for finding this error. It is now corrected.*

page 9, line 8: This is a slant path for the integration; make more clear.

*Text was changed to "...is the integral of the trace gas concentration along the slant path between the instrument and the sun..." to clarify this.*

page 13, line 18: The albedo of ocean (0.2 in visible) seems high. Why is this chosen? Reference?

*The following explanation was added to clarify the origin of the albedo values: "...albedo is set to 0.07 in the UV, and 0.2 in the VIS, to account for the broken cloud cover observed by the on-board camera and the CPL, i.e. the albedo values are a composite of pure ocean albedo and cloud albedo."*

page 16, line 3: Replace "it" with "O4 reference", and indicate "for general (e.g. not fully clear sky) conditions."

*Changed sentence to: "This renders $O_4$ absorptions unsuitable for the interpretation of high-altitude aircraft DOAS applications for all sky conditions (e.g. without a fully clear sky over the large area to which the limb observations are sensitive)."*

page 18, line 13: There are three citations to theses in this line, and in general these theses may not always be accessible, nor are they fully peer reviewed. Other instances of theses also exist in the citations. It should be made clear that this method is de-scribed in those theses, but was not published in peer-reviewed form in those theses. See AMT reference guidelines.

*We do not fully agree with this argument. PhD theses are often better reviewed than books and reports cited in peer-reviewed literature. Historically, theses have also been considered publications. We acknowledge that finding a thesis may be more challenging, and have therefore added the links to the Univ. Heidelberg thesis repository to the respective references.*

page 20, line 33: Please also show the ratio of alpha factors to demonstrate the degree of "cancellation" of the variability of the individual factors.

*We added the ratio to the figure and an additional sentence to the caption: "Panel c shows the ratio of the α-factors in each wavelength range."*

*We also added a sentence in Section 4.3 explaining that the α-factors ratio depends on altitude: "It should be noted that the $\alpha_{Xj}\backslash\alpha_{Pj}$ ratio strongly depends on the relative profile shapes of the involved gases, and thus altitude"*

page 22, line 7: I believe the column of overhead ozone is varied, not the shape of the profile. Please clarify.

*Correct, the text was changed to "…$O_3$ column density…"*

**Response to Reviewer 2:**

Most aspects are well explained and the quality of the measurements and analysis is good. However, the text would benefit in some parts if the authors put some more carefulness in the writing, i.e. the phrasing and correct grammar. There are quite a number of mistakes/oversights such as missing words, as well as a few passages which are not so well described. This makes the reading somewhat cumbersome in the respective places, while the content itself is mostly fine. Please check the text once more in addition to the comments given here. With some additional effort on the text and some clarifications, the manuscript is certainly suitable for publication in AMT.

I.      Main Comments

Comment 1. Section 4.2, l. 1059
Please describe and/or cite what is used as "typical BrO concentration profile". It is also unclear how the choice of this influences the uncertainty determination.

Page 17 line 23:
*We replaced the text to clarify this: "… a BrO concentration profile based on observations during the second profile maneuver of the ATTREX flight on 5–6 February 2013 …" This section exemplifies the sensitivity of the retrieval to the presence of lower atmospheric clouds to motivate the development of the scaling technique. Since we are not using the optimal estimation technique for the rest of the manuscript, a more detailed discussion of the optimal estimation uncertainties, for example due to the BrO profile shape, is beyond the scope of this manuscript.*

Comment 2. Section 4.3, l. 1202-1208
Please give some more clear explanation on how (strong) the model assumptions and especially the finally applied values of SCDref and the vertical trace gas profiles from the TOMCAT/SLIMCAT model, which constrain the RT simulations, influence the mixing ratio results received from the O3-scaling method.
This influence is also mentioned further below (Section 4.3, l. 1242). It becomes clear that certain effects and assumptions cancel in the applied method but others don't. It is also clear from the detailed descriptions and equations that this input information is needed. However, it is necessary to understand in how far the results and resulting interpretations from the announced new scaling method are independent from the initial assumptions, especially SCDref and the vertical profiles.
The Supplement gives a wide range of valuable information on influences from the
model assumptions on the final results, as summarized in Table 5. However, SCDref is not considered and the vertical profiles are only scaled (either for the tropospheric or the stratospheric part) but not changed in their actual shape. If this (i.e. absolute changes as opposed to the shape in the two separate altitude ranges) is representative for the entire profile influence on the final results this should be stated more clearly. One

could imagine that especially relative changes in the profile shapes between ozone and NO2 or BrO are relevant for the final mixing ratio results. Please add some explanations as clarification.

*The shape of the trace gas profiles entering $SCD_{ref}$ is unimportant in our case due to the use of diffusors in our instrument. Consequently, the absorption path is simply the geometric path between the instrument and the sun. For such a geometric path only the vertically integrated column above the aircraft matters, not the actual shape of the trace gas profile above the aircraft. We have revised the text in Section 4.3 (page 20, line 21 - 25) to clarify this:*

*"...$SCD_{ref}$ calculated from the Tomcat/Slimcat model results of the vertical trace gas column above the aircraft and the geometric airmass factor for a direct sun observation. It should be noted that the use of direct sun observations makes this calculation independent of the shape of the overhead trace gas profile and highly simplifies the radiative transfer. Direct sun $SCD_{ref}$ observations thus provide an advantage over the commonly used scattered light solar references, for which both overhead profile shape and much more complicated radiative transfer impact $SCD_{ref}$.*

*To further clarify this point we revised the respective text in Section 4.4 to state:*

*"However, for small $X_j$ the errors due to uncertainties of $\Delta SCD_{ref}$, i.e. from the vertical trace gas column above the aircraft, have to be considered (Runs 3 to 6, Figures 2 and 11 in the supplement). Because we use direct sun observations, changes in the overhead profile shape do not impact $SCD_{ref}$ or its uncertainty."*

Comment 3. Section 4.3, ll. 1269-1303, Figures 12 and 13
In part this comment is connected to Comment 2. It is not clear how the shifting procedure is performed. At first reading the procedure seems easy. However, comparing with Figures 12 and 13 some questions arise. First of all, the caption of Figure 13 is really misleading. It states that the measurement results are forced to match the model results, which according to text and figures should be vice versa. The measurement curves are the same as in Fig. 12, and the model results change from Fig. 12 to Fig. 13. As stated in the text, the model is forced to match the measurements, correct?

The change in ozone between pre and post shifting in some parts of the flight, however, is quite large up to a factor of two it seems (e.g. between 18:00 and 20:00 UTC), and maybe around 10-20% in other parts of the flight. The text, however, states that the vertical shift needed for matching the curves is usually smaller than 1km. Is the ozone changing that strongly between one model layer and the next? Maybe there is some misunderstanding. Maybe it would be also useful to give some more information on the used vertical profiles in general, e.g. in the Supplement, see also Comment 1. In addition: Why is a vertical shift sufficient? Is this simply because of monotonically increasing concentrations? Please briefly clarify.

*Thanks for pointing this out. The caption in Figure 13 was stating the opposite of what we actually did. This mistake has now been corrected and the caption now states:*
*"Same as Figure 12, but vertically adjusting TomcatT/Slimcat-modelled predictions with NOAA-measured $O_3$ to agree in a least square sense."*

*At the beginning of the flight, when the Global Hawk was flying at the lower border of the stratosphere, trace gases do indeed show large vertical gradients. This can clearly be seen in Figure 6, which shows the trace gas curtains and the flight altitude. We recognize that we did not discuss the consequences of the vertical adjustment for the first few hours of the flight and revised the last paragraph of Section 4.3 to better explain this:*

*"The impact of this sub-grid scale correction of the model results on BrO and $NO_2$ concentrations is illustrated in Figure 12 and Figure 13. The vertical adjustment leads to larger changes in $O_3$, and*

*consequently CH$_4$ and NO$_2$, before 23:00, as the GH was flying on the lower edge of the stratosphere, where large vertical gradients of these gases can be found (see also Figure 6). Even small altitude changes can therefore lead to a substantial change in the concentrations. The concentration changes due to the vertical adjustment are smaller after 23:00, when the GH was flying in the UTLS and vertical concentration gradients are smaller (Figure 6). Retrieved BrO mixing ratios generally show a smaller change as a consequence of the vertical adjustments, due to the smaller BrO gradient during the earlier part of the flight in the LS, as well as during the second half of the flight in the UT."*

Comment 4. Section 6
As outlook and for a better understanding of how the measurements performed by the new DOAS instrument on the UAS will yield further information on the UTLS, it would be very helpful if a statement - e.g. at the end of the conclusions - would be given on the future plans, especially on how often and for how long etc. the instrument can - and will be - used for the type of measurements presented here, and for which further research the results will be helpful.

*We revised the last paragraph of our manuscript as follows to address this useful comment:*
*"Our companion paper (Werner et al., 2016) provides a quantitative analysis of UTLS bromine chemistry for the entire ATTREX 2013 data-set derived by the methods described here. Observations from the ATTREX 2014 deployment are also undergoing analysis with respect to the chemistry in the western-pacific UTLS. The methods and results presented here provide guidance for the design of future high-altitude airborne experiments to study the composition and chemistry in the UTLS."*

II.        Minor Comments

1.        Lines 19-20: It is somewhat strange to concentrate on relative errors here. In order to judge the performance of the instrument, absolute errors are in addition helpful.

*These lines refer solely to the DOAS retrieval errors, which is only one part of the overall uncertainties of the final trace gas mixing ratios. Because we employ a direct sun solar reference spectrum, the BrO mixing ratios are higher than in the typical case of using a scattered light solar reference. This leads to a somewhat higher absolute, but not relative, error of the DOAS retrieval. We have therefore decided to leave these lines unchanged.*

2.        Line 57: Please state and cite actual values (current best knowledge) of total amounts of Brorg and Br so the given numbers can be directly set into relation.

*We added a statement of the total inorganic bromine in the stratosphere taken from the 2014 WMO report: "…presently assessed at 16 - 23 ppt (WMO, 2015)… ". We also added a parenthesis with the total long-lived organic bromine species to the text: "(15.2 ±0.2 ppt in the troposphere (WMO, 2015))". We believe this should give the reader sufficient information to put the other numbers in this section into perspective, and also provides a reference to the latest WMO report summarizing our current knowledge.*

3.        Line 87: Is 3ppt the current uncertainty or the goal for the improved uncertainty of Bry? Please state the respective other quantity. (The newly achieved uncertainty would be interesting, however, according to the grammar of the sentence the given value is the current uncertainty. Please clarify.)

*This is indeed the current uncertainty. To make sure the reader understands where this uncertainty comes from we added a reference to the 2014 WMO report. Ultimately one would always want the desired uncertainty to be as small as possible. However, the uncertainty that can be achieved is determined by that*

*of the observations/calculations. Therefore, it seems best to state that the goal is to decrease the uncertainty below the current uncertainty.*

4.        Line 97: Earliest measurements of? Does this refer to BrO?

*Yes. Sentence changed to: "The earliest BrO measurements were performed using in-situ chemical conversion resonance fluorescence instruments…"*

5.        Lines 305-307: Please give a short reason (too dark due to lack of scattered sunlight?).

*" … due to the low intensity of scattered sunlight." Was added to this sentence*

6.        Lines 388-390: This does not really explain how the opening angles are actually determined. The scanning capability obviously helps in the calibration. Please shortly state how the FOV was measured.

*This is explained two paragraphs below: We added "…, as explained further below" to refer the reader to the following sentence: "The elevation opening viewing angle is also accurately determined by scanning the telescopes over the light beam, while recording the light spectral radiance and pointing elevation angle at the same time.", to which we also added that the pointing elevation angle is also recorded.*

7.        Lines 425-428: Although this range of the telescope stability is within the uncertainty of the elevation calibration, it obviously adds to the overall uncertainty of the determined elevation. Is this included in the uncertainty considerations?  It doesn't sound as if it is. Or maybe the phrasing is misleading?

*The wording was misleading; this is indeed the overall uncertainty. We removed the last part of the sentence ("…, which is within the uncertainty of the elevation calibration") to remove any misunderstanding.*

8.        Lines 457-461: What about the roll angle? Probably it is just missing in the list.

*We added roll and heading to this list.*

9.        Lines 549-551: A common shift would not be an error of the trace gas reference spectra which are all from individual data, but a (very small in the presented case) uncertainty from the instrument calibration which seems to be fine.

*Sentence changed to "Inaccuracies in wavelength position due to small changes in the instrument during the observations, and errors in the instrument wavelength calibration used to calculate the reference spectra, are corrected during the spectral retrieval procedure."*

10.       Line 556: Please refer to Table 3.

*done*

11.       Lines 567-570: Reading this remark, the reader asks for an explanation, which is only given much later. Either shortly state or refer to Section 4.

*A reference to Section 4 was added*

12.     Lines 575-578: The excluded range lies within the fitting windows for O3 and BrO also, but is not excluded there, right? A short comment of reasoning would be helpful. (Why does the feature lead to a problem only in the O4 fit?)

*To retrieve $O_4$ accurately, a larger wavelength interval with a low order polynomial has to be used. This is likely the reason why this feature does not cause any problems in the $O_3$ and BrO window. The reason for the feature is unclear. The manuscript already includes an explanation for this:*
*"Due to the use of a low polynomial degree of 2, the spectral range between 347.4 – 352 nm shows an unidentified residual structure. Therefore this wavelength interval is excluded from the spectral retrieval. The use of a higher degree polynomial decreases this residual structure, but causes the $O_4$ retrieval to be less stable."*

13.     Line 608: Please give the ozone window also in the text as for the other trace gases, not only in the table.

*done*

14.     Line 644: New paragraph?

*Thanks for pointing this out. This was a mistake in the original TEX file.*

15.     Lines 698-705: Please check this passage, there is some inconsistency. If the reduction in O3 and NO2 is "not as pronounced" it would mean that at EA 1∘ the DSCD is similar as in the lower EA, so the relative error for O3 and NO2 would be even more problematic if using EA 1∘ as reference instead of direct sun. Please clarify.

*"pronounced" was not a good choice of word. We have reformulated the sentence as follows to clarify this point: "While this reduction is not as important for $O_3$ and $NO_2$, because the relative DSCD errors are generally small, it is more serious for BrO, where the change of the DSCD with EA is only half of the overall signal and the relative error is larger."*

16.     Lines 749 and 763: Is it true that one instrument/method gives relative errors and the other one absolute errors only?     Otherwise please use comparable data/information.

*We added the respective information on the precision in parentheses at both places in the text.*

17.     Line 792: Is an albedo of 0.2 realistic for the VIS? This seems rather large.

*The following explanation was added to clarify the origin of the albedo values: "…albedo is set to 0.07 in the UV, and 0.2 in the VIS, to account for the broken cloud cover observed by the on-board camera and the CPL, i.e. the albedo values are a composite of pure ocean albedo and cloud albedo."*

18.     Lines 845-849: Please briefly state which observation leads to the interpretation (transition from lower stratosphere to UTLS region).

*It was the DSCD observations of $O_3$, and $NO_2$ shown in Figure 4. The trace gases were added in this sentence.*

19.     Lines 1006-1007: This information should be given earlier in the section, i.e. at first mentioning.

*Thanks for pointing this out. The sentence was moved directly after equation (3).*

20.    Lines 1389-1391:  As the sentence can be read in a somewhat misleading way, a recommendation would be a rephrasing of the following kind: "As further analysis goes beyond the scope of this manuscript, we only briefly discuss the results (...)."

*done*

21.    Lines 1485-1486: (...) i.e. low mixing ratios in  the  troposphere  (...).  This is, however, not generally true for NO2 at all. This statement requires some clarification.

*The only exception to this statement seems to be highly polluted areas, where boundary layer $NO_2$ mixing ratios can exceed those in the stratosphere. We have thus changed the sentence to:*
*"Because ozone, BrO, and $NO_2$ have similar mixing ratio profiles in the unpolluted atmosphere, i.e. low mixing ratios in the troposphere and high mixing ratios in the stratosphere, most of the RT effects and uncertainties cancel out in the $O_3$-scaling technique."*

22.    Lines 1216-1219 and Figure 10:  Does the calculation depend on certain experimental flight conditions? Just trying to understand why there are kinks in the curve, e.g. at 700s. This is not clear. Please give a short explanation.

*This curve was determined for typical flight conditions. The small kinks are due to the statistical nature of the Monte-Carlo model, i.e. they are a reflection of the uncertainty of the simulation of this curve. The caption was updated, and now reads:*
*"Relative contribution to the analyzed light as function of the time ahead of the actual flight position, as predicted by 3D simulation of the radiative transfer for typical flight conditions (Raecke, 2013). The small random variations past 600 s are due to the statistical nature of the Monte Carlo RTM, which introduce random uncertainties"*

1.    Line 5-9: (...) developed for (...) GH (...) _and tested/applied_ during (...) ATTREX.
      *Changed to "(GH) and deployed during the Airborne Tropical TRopopause EXperiment (ATTREX)"*
2.    Line 26:  (RT) not needed in abstract,  abbreviation should be given in main text   (Line 199?).
      *done*
3.    Line 151: (...) interpretation _of_ the data.
      *already fixed in published AMTD version*
4.    Line 159: Sounds as if the name of the aircraft is missing.
      *removed a "the" to fix this error.*
5.    Line 181: New paragraph?
      *done*
6.    Line 193: Please add abbreviation DSCD at the end of the sentence as definition.
      *done*
7.    (...) the side facing _of_ the diffusers (...)
      *already fixed in published AMTD version*
8.    Line 541: The term "dispersion" seems to be the wrong choice. Do you rather mean "convolution"?
      *No, dispersion is correct, the dispersion is used to map wavelengths to pixel.*
9.    Line 673: (...) and in _the/a_ companion paper (...)
      *Fixed.*
10.   Lines 770-773:  Sentence  misses a verb.       E.g.     (...) we  also  employ a 3D atmospheric chemistry model to _determine_ vertical trace gas concentration profiles.
      *Fixed.*

11. Lines 813-816: Although (probably) not wrong, the sentence structure makes it really hard to understand the meaning, especially the subject of the sentence "TOMCAT/SLIMCAT model meteorology".
*We deleted the first part of this sentence to make it more understandable. The sentence now reads: "Tomcat/Slimcat model meteorology, i.e. large-scale winds, temperatures, as well as convective mass fluxes, is driven by ECMWF ERA-interim reanalyses."*

12. Line 832: Remove the space at the end of the sentence.
*We removed a space in line 828, but could not find an extra space in line 832.*

13. Lines 860-861: (...) at the time and location _of_ the measurement (...).
*done*

14. Lines 987-989: Missing word "and": (...) _and_ a non-linear retrieval (...) is employed.
*done*

15. Line 996: The solution of this iteration is _a_ (instead of "an") vertical (...) profile (...).
*done*

16. Line 1052: (...) the general error of _a_ (instead of "an") non-linear extinction (...)
*done*

17. Lines 1077-1082: Devide into two sentences please.
*The two sentences now read: "However, for the GH altitude and a limb observational strategy, the uncertainty of low cloud OD and/or the possibility of an OD change during an ascent or descent can introduce significant errors in the retrieved BrO concentrations.*
*These errors are typically larger than those needed for the investigation of UTLS bromine chemistry..."*

18. Line 1081: "than" instead of "that"
*done*

19. Line 1124: typo in "predomina_n_tely"
*done*

20. Line 1125: (...) the ratio of the two parameters _is_ (instead of "are") a measure (...).
*done*

21. Line 1140-1142: Please correct this sentence.
done

22. Line 1144: Due _to_ (...)
*done*

23. Line 1171: "B" should be in italic format.
*done*

24. Lines 1269-1270: (...) requires addition_al_ scrutiny (...)
*done*

25. Line 1457: Please delete the extra "the".
*done*

26. Line 1460-1461: Apparently this sentence belongs to a different paragraph. It does not fit to the content here at all, but probably should rather be shifted to line 1490.
*The purpose of the sentence was to point out the usefulness of direct sun reference measurements. The word 'scaling approach' was a mistake here and has been corrected.*

27. Lines 1470-1471: The verb "are" needs to be repeated here at least once: (...) and _are_ optically more powerful (...).
*We added a verb and also split the sentence in two to make it more understandable.*

28. Line 1481: (...) needed to be develop_ed_ (...).29. Table 2: The wavelength range for the VIS instrument really seems to be wrong. The given range from 410-425 nm would not cover the retrieval windows given in Table 4.
*Corrected to 410 - 525 nm.*

30. Table 4: Please correct the title of the last column. e.g. Average DSCD error (molec/cm2, for O4: molec2/cm5) or similar.

*done*

31.      Figure 2 and 3: O3 (please use subscript).
*done*

32.      Caption of Figure 2: delete second "the" in second of last line.
*done*

Missing plural "s", e.g., at the following locations: Line 153: Few research aircraft_s_
Line 158: (...) a number of aircraft_s_
*The plural of 'aircraft' is 'aircraft' no '-s' is needed at the end*

Line 214: Theoretical consideration_s_
*done*

Line 1046: (...) these result_s_
*done*

Line 1290: (...) the modeled O3 profile_s_ are (...)
*done*

Line 1326: (...) spectral retrieval error_s_ (...) which are (...) Line 1471: (...) instrument_s_
*done*

Corrections w.r.t. references and citations:

1.      Line 40 (and many other locations): The format of references / citations has some issues. Please remove the extra parentheses around the year of publication.
*done*

2.      Line 163: Parentheses around the two references are missing.
*done*

3.      Please correct typo in name Baidar in Baidar et al., 2013.
*done*

Corrections concerning the Supplement:
Text correction in the figure captions: (...) parameters given _in_ Table 5.
*done*